# Inter-Comparison of Landsat-8 and Landsat-9 during On-Orbit Initialization and Verification (OIV) Using Extended Pseudo Invariant Calibration Sites (EPICS): Advanced Methods

Morakot Kaewmanee, Larry Leigh *, Ramita Shah and Garrison Gross

Image Processing Lab, Department of Electrical Engineering and Computer Sciences, South Dakota State University (SDSU), Brookings, SD 57005, USA; morakot.kaewmanee@sdstate.edu (M.K.); ramita.shah@jacks.sdstate.edu (R.S.)
* Correspondence: larry.leigh@sdstate.edu; Tel.: +1-605-688-5918

**Abstract:** Three advanced methodologies were performed during Landsat-9 on orbit and initialization and verification (OIV): Extended Pseudo Invariant Calibration Sites Absolute Calibration Model Double Ratio (ExPAC Double Ratio) and Extended Pseudo Invariant Calibration Sites (EPICS)-based cross-calibration utilizing stable regions in Northern African desert sites (EPICS-NA) and a global scale (EPICS-Global). The development of these three techniques was described using uncertainties analysis. The ExPAC Double Ratio was derived based on the ratio between ExPAC model prediction and satellite measurements for Landsat-8 and Landsat-9. The ExPAC Double Ratio can be performed to determine differences between sensors ranging from visible, red edge, near-infrared, to short-wave infrared wavelengths. The ExPAC Double Ratio and EPICS-based inter-comparison ratio uncertainties were determined using the Monte Carlo Simulation. It was found that the uncertainty levels of 1–2% can be achieved. The EPICS-based cross-calibration results were derived using two targets: EPICS-NA and EPICS-Global, with uncertainties of 1–2.2% for all spectral bands. The inter-comparison results between Landsat-9 and Landsat-8 during the OIV period using the three advanced methods were well within 0.5% for all spectral bands except for the green band, which showed sub 1% agreement.

**Keywords:** Landsat-9; Landsat-8; EPICS; absolute calibration model; inter comparison; ExPAC model; ExPAC Double Ratio; EPICS-based cross-calibration; double ratio; hyperspectral BRDF coefficients

## 1. Introduction

In recent years, the Extended Pseudo Invariant Calibration Site (EPICS) has been used for radiometric calibration since its introduction in 2018. The coverage of stable pixels in Northern African desert sites provided temporal stability at levels of 3–4% for all Landsat spectral bands and high temporal resolution of image acquisition, every 1–2 days, for satellite calibration purposes [1–3].

The EPICS major advantages over traditional single PICS (Pseudo Invariant Calibration Site) are firstly, temporal data acquisition is increased every 1.5 days versus every 16 days over a single target under the ideal conditions of cloud-free images. Secondly, the EPICS has reduced the impact of individual inherent site anomalies by expanding stable sites to a continental level [3] and on a global scale [4]. Thirdly, it has the ability to increase radiometric assessment at a finer timescale, especially short-lived sensors. Lastly, the EPICS and PICS are proven to provide an alternative technique for radiometric calibration to substitute on-board calibrators as they are inexpensive and reliable calibration approaches [5–10].

Traditionally, on-orbit and initialization and verification (OIV) took several months to confirm that the newly launched satellite was ready for routine operation. EPICS has been utilized to perform OIV test for cross-calibration between Landsat-8 and Landsat-9 which led to assisting the decision-making to update CPF (Calibrated Parameter File) gains update

within 3 months in orbit. It was also confirmed that the agreement between Landsat-8 and Landsat-9 are at 0.5% level for all bands except the green band with sub 1% level with six independent techniques for validation: Underfly event, EPICS Trend to Trend (EPICS-NA, EPICS-Global), EPICS-based cross-calibration (EPICS-NA, EPICS-Global) and Extended PICS Absolute Calibration Double Ratio (ExPAC Double Ratio) [11].

This paper focused on three advanced methods for cross-calibration: Extended PICS-based cross-calibration for EPICS-NA and EPICS-Global and Extended PICS Absolute Calibration Double Ratio (ExPAC Double Ratio). The development of these three advanced techniques will be described in the following sections and followed by uncertainties analysis and results and discussions. The final section will be a summary and conclusions.

### 1.1. Extended Pseudo Invariant Calibration Site in North African Desert Sites (EPICS-NA) and a Global Scale (EPICS-Global)

The study of finding stable pixels for satellite stability and radiometric assessment has been actively reported using northern Saharan desert sites over the past two decades [12–15]. An algorithm to identify an optimal region exhibiting 3% or less spatial, temporal, and spectral uncertainties over Libya-4, Libya-1, Niger-1, Niger-2, Sudan-1, and Egypt-1 was developed at South Dakota State University, Image Processing Laboratory—SDSU IPLab in 2017 [5,16]. When these optimal regions were used to perform lifetime trending for Landsat-8, the final weighted average drift estimates across the six PICS could achieve the same results as on-board calibrator with less than 0.5% difference; moreover, the temporal uncertainty of Libya-4 was well below 1.5% for all spectral bands. For all other 5 PICS, the temporal stability was well below 2% [3,17,18].

Shrestha et al. [19] developed an algorithm to classify the sand types using Landsat-8 data in the North African region, using a threshold of 5% temporal and spatial uncertainties. The classification algorithm identified 19 Clusters which represent 19 distinct surface types. Among these clusters, the largest cluster which extended across North Africa and possessed a reflectance similar to Libya-4 was Cluster13, which for this study it will be referred to as EPICS-NA, allowing the opportunity to acquire a calibration point more frequent to nearly a daily basis which is much better than the conventional every 16 days revisit time for a single location. The EPICS-NA stable pixels were shown in red with Landsat-8 images as seen in Figure 1. The temporal stability of EPICS-NA was around 1.2–2.4% for green, red, NIR, SWIR1 and SWIR2 and ~3% for Coastal aerosol and the blue band [2]. This finding has extended the capability of utilizing PICS for the potential of having the largest region for absolute calibration with high temporal resolution.

Fajardo Rueda et al. [4] further expanded the classification to a global scale, modifying the K-mean algorithm without any thresholds. In total, 160 classes were assigned in the classification process. Using the same criteria to determine a class that is spectrally matched Libya-4, these stable pixels were named 'EPICS-Global'. The temporal stability was ~2.8–3.2% for all bands except for red and SWIR-2 bands, which are 3.8% and 4.6%, respectively. Furthermore, the refinement of global classification was carried out setting the total classes to 300 with same criteria. The EPICS-Global with 300 classes was stable within 2.2–3.8% for all bands. The Landsat-8 images over 33 Path(s)/Row(s) represent EPICS-Global around the world, providing a daily calibration opportunity [20] as shown in Figure A1.

### 1.2. Extended PICS Absolute Calibration Model (ExPAC Model)

Many researchers have used PICS to develop an absolute calibration model. Govaerts et al. initiated work in developing an absolute calibration model using Libya-4 PICS as a target. The main concept of their work was to apply the relative trending curve of the tar-get spectral profile of any sensor using PICS and anchoring it to an absolute calibration of a reference sensor [4]. The accuracy of the model was within 3% for red and near infrared bands. In 2012, the absolute calibration model was improved with the use of

an advanced radiative transfer model in cooperating radiation polarization to improve surface characterization [5].

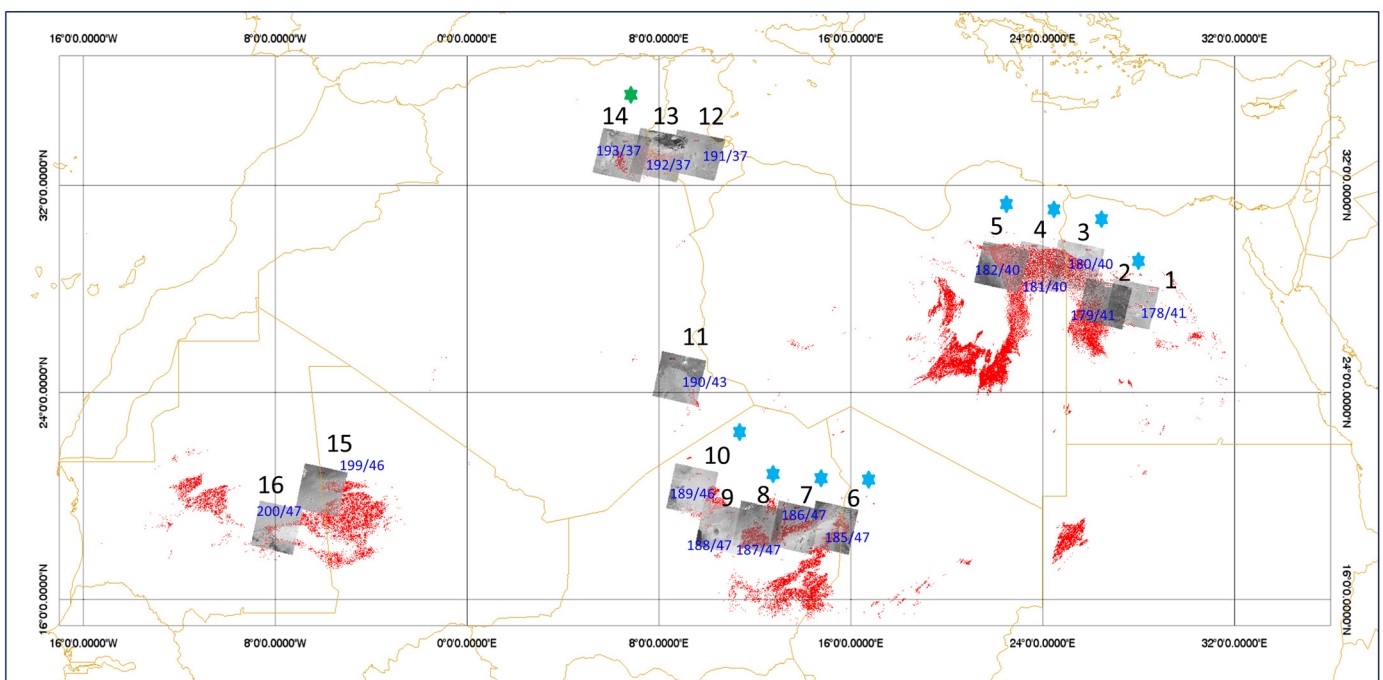

**Figure 1.** EPICS-NA stable pixels shown in red, 16 Path/Row Landsat-8 image footprints, the asterisks denote Path/Row(s) selected for Landsat-7 data used in the validation process.

South Dakota State University's Image Processing Laboratory has had a long history using the PICS absolute calibration models since 2012. It was developed with Landsat-7 Collection-0 data using Libya-4 PICS as a target; an empirical absolute pseudo-invariant calibration model was called Libya-4 APICS model using Terra MODIS as a reference radiometer [21]. It should be noted that this empirical absolute calibration model was based on solar zenith and viewing angles in spherical degrees. In 2017, the Libya-4 APICS model was improved to include a coastal aerosol band using Landsat-8 data, and was named a Refined APICS model. The adjustment of K factor–scaling factor to place derived TOA reflectance from Hyperion to MODIS calibrated scale was replaced to accommodate the coastal aerosol band. The overall model accuracy of this refined APICS model was well below 3% for all bands [22]. This absolute calibration technique had been extended to five other PICS sites: Libya-1, Niger-1, Niger-2, Egypt-1, and Sudan-1, showing a comparable model accuracy of 3% for all bands and all sites except Niger-2, which had 2–7% accuracy [9].

Since 2017, the Landsat image archive has been reprocessed into Collection-1 data, consisting of Level 1 products that meet formal geometric and radiometric quality criteria: better than 12 m RMSE and better than 3% uncertainty, respectively [23]. It also provides solar and sensor angle information and quality assessment bands. Farhad et al. [24] had a new BRDF model developed using solar and satellite illumination angles taking advantage of Landsat-8 Collection-1 data, which have provided information on the sensor and solar geometry angles since 2018. Instead of using angles in spherical degree units to develop a 4-Angle BRDF model with solar zenith, solar azimuth, satellite view angle and satellite azimuth angle, these angles were converted into Cartesian Coordinates. Then, this 4-Angle BRDF model was derived using a quadratic fit and interaction terms. When the 4-Angle BRDF model was applied to Landsat-8 Libya-4 PICS, the temporal variation for all spectral bands was well within 1.5%. In longer wavelengths, a major improvement was better than 10% when compared with BRDF normalization using a simple linear BRDF solar zenith model [25]. These findings ignited the concept of utilizing the quadratic and

linear relationships between these four angles to improve an empirical absolute calibration model.

Das Chaity et al. [10] developed a hyperspectral empirical absolute calibration model using EO1-Hyperion data to generate a 4-Angle BRDF model using angles in Cartesian Coordinates instead of spherical angles. The model depicted a quadratic fit for illumination angles and a linear model for satellite viewing angles. The hyperspectral absolute calibration model performed well within 3% accuracy when solar zenith angles and solar azimuth angles are 20 to 50 degrees and 96 to 260 degrees, satellite viewing angles are within 17 degrees, and satellite azimuth angles are 80 to 292 degrees.

The development of an absolute calibration model with illumination angles and satellite viewing angles was initiated using EPICS-NA as a target and Landsat-8 as a reference sensor in 2019. It was named Extended PICS Absolute Calibration Model or 'ExPAC model'. Using Landsat-8 Collection-1 data, the initial empirical absolute calibration model was a simple quadratic model of $X_1$ and $Y_1$ parameters as expressed in Equations (1) and (2) [18]. The ExPAC Model was validated with Landsat-7, Landsat-8, Sentinel-2A, and Sentinel-2B; showing agreement between the model predicted and measurements at 2% level of accuracy. However, there was an obvious limitation of the BRDF model for Sentinel's wider field of view which could be improved [18].

This study presents the improvement of the ExPAC model, a combination of data from three satellites: Landsat-8 Collection-2, Hyperion and MODIS Terra was constructed to be spectrally matched with Landsat-8, for this study it was referred to as 'Harmonized data'. It was used to generate a BRDF model that can accommodate larger viewing angles and solar illumination angles. Thus, a new ExPAC model would be developed. Details will be discussed in the next section.

## 2. The Development of Extended PICS Absolute Calibration Model (ExPAC Model)
### 2.1. Data

The Extended PICS Absolute Calibration Model (ExPAC model) was developed using Landsat-8 as a reference radiometer. EO-1 Hyperion and MODIS Terra were used to extend solar and sensor viewing angular variation. Landsat-8 Collection-2 data, MODIS Terra, and Hyperion data were harmonized to generate a dataset for further BRDF model generation. Once the ExPAC model was developed, it would be validated using Landsat-7 ETM+, Landsat-8 OLI, Sentinel-2A MSI (S2A), Sentinel-2B MSI (S2B), MODIS Terra, and MODIS Aqua. The summary of data properties in developing the ExPAC model was shown in Table 1.

**Table 1.** Shows the details of data used in developing the ExPAC model data from Beginning of life until end of January 2021. Information on all the angles was extracted from the angle information band and metadata.

| Satellite Names | Launch Date | VZA | VAA | SZA | SAA | No. Scenes | Remarks |
|---|---|---|---|---|---|---|---|
| Landsat-8 | 11 February 2013 | 3° to 8° | 90° to 98° 270° to 280° | 20° to 60° | 73° to 160° | 2741 | 16 sites |
| Landsat-7 | 15 April 1999 | 3° to 8° | 90° to 98° 270° to 298° | 19° to 58° | 73° to 158° | 1952 | 9 sites |
| Sentinel-2A | 23 June 2015 | 2° to 12° | 90° to 97° 270° to 296° | 15° to 59° | 71° to 167° | 3583 | 16 sites |
| Sentinel-2B | 7 Mar 2017 | 2° to 12° | 90° to 98° 270° to 291° | 15° to 9° | 71° to 167° | 1784 | 16 sites |
| MODIS Terra | 18 December 1999 | 1° to 30° | 90° to 98° 270° to 291° | 12° to 55° | 81° to 167° | 1285 | 1 site-Libya-4 |
| MODIS Aqua | 4 May 2002 | 1° to 9° | 200° to 260° | 16° to 56° | 79° to 83° 90° to 102° | 1573 | 1 site-Libya-4 |
| EO-1 Hyperion | 21 November 2000 | 0° to 25° | 90° to 98° 270° to 291° | 21° to 77° | 70° to160° | 667 | 16 Sites |

### 2.1.1. Landsat-7, 8, 9

Landsat-7 was launched into orbit on 15 April 1999, and its calibration uncertainty is better than 5% at the top of atmosphere (TOA reflectance) for all six spectral bands: visible, near-infrared (NIR) and short-wave infrared (SWIR) bands [6]. Landsat-7's orbit was lowered two months prior. Landsat-9 launched to allow Landsat-9 to take its orbital place. In mid-2022, Landsat-7's crossing equator time was 9.00 am local time [26].

Landsat-8 was launched into orbit since 12 February 2013 and its calibration uncertainty is better than 3% in top-of-atmosphere (TOA) reflectance for all seven spectral bands: Coastal Aerosol, Visible, NIR, and SWIR bands [27]. Since 2017, the Landsat image archive has been reprocessed into Collection-1 data, which consist of Level 1 products that meet formal geometric and radiometric quality criteria: better than 12 m RMSE and better than 3% uncertainty [6]. It also provides angle information and quality assessment bands. The Landsat-8 Collection-2 Level-1 data were introduced in early 2021, and its major improvement in geometric accuracy is comparable to Europe's Copernicus Sentinel-2 missions. The radiometric calibration update was applied to the Coastal Aerosol and blue bands [28]. They can be publicly accessed and downloaded from www.earthexplorer.usgs.gov.

Landsat-9 was recently launched on 27 September 2021. It inherits identical spectral bands with four visible spectral bands, one near-infrared and two shortwave-infrared bands at 30 m. spatial resolution. Unlike Landsat-8, with radiometric resolution of 12 bit, Landsat-9 was designed to have 14 bits resolution [29]. Landsat-8 and Landsat-9 are orbiting in constellation with repeat coverage over the same location at 8 days cycle.

The digital number is converted to be TOA reflectance as detailed in [23]. For Landsat, the 4-Angle information was extracted from the angle bands. The satellite data in Collection-2 were used from the first available date in the archive until mid-2021 for all 16 paths/rows of EPICS-NA, as seen in Figure 1. Only Landsat-7 images with 9 Paths/Rows (depicted in blue and green asterisks) were downloaded to minimize the number of datasets when processing the data for EPICS-NA since 1999, as shown in Figure 1. The Landsat data were filtered using Band Quality Assessment and visual inspection, ensuring that all selected pixels were cloud-free.

### 2.1.2. EO-1 Hyperion

EO-1 Hyperion was launched into orbit on 21 November 2000. It was a hyperspectral sensor providing 196 bands ranging from 400 nm to 2500 nm, at 10 nm spectral resolution and 30 m spatial resolution. Hyperion was a pointing satellite up to $\pm25\,^\circ$ off-nadir orientation. The latest radiometric calibration study was carried out in 2017, showing the calibration uncertainty well within 5% for the VNIR and within 10% for the SWIR bands for 16 years in orbit [30]. The Hyperion data were acquired using the information of EPICS-NA, where 667 scenes were collected. When the Hyperion sensor was decommissioned on 20 March 2017, there was an analysis of lifetime absolute calibration using Libya-4 PICS data to calculate yearly drift and determine the absolute radiometric calibration of the sensor via vicarious reflectance-based calibrations performed at the South Dakota State University (SDSU) test site and the Radiometric Calibration Network (RadCalNet) Railroad Valley site using data from 2002 to 2015. The results were validated with Landsat-7 data showing that after applying significant drift and calibration coefficient correction, there is no significant gain and bias in all test sites: Lake Tahoe, Railroad Valley, Amazon Forest, White City, and Libya-4 [31]. Hence, the Hyperion data were corrected for yearly drift and absolute calibration gain as described in [31] before being used in combination with Landsat-8 data.

### 2.1.3. Sentinel-2A, Sentinel-2B (S2A, S2B)

Sentinel-2A and Sentinel-2B were launched into orbit on 23 June 2015, and 7 March 2017, respectively. The MultiSpectral Imager (MSI) is a push-broom imager with 13 spectral bands from VNIR to SWIR. It consists of 12 individual arrays of detectors covering 290 km swath width. The spatial resolution of Sentinel-2A and 2B is 10, 20, and 60 m dependent

on the particular band. The repeat cycle is every 10 days, with a constellation, the repeat coverage is every 5 days for these two satellites. The absolute calibration of all 13 bands has been reported to be better than 5% (target at 3%) at Top of Atmosphere level [32]. There have been several processing versions ongoing throughout the lifetime of Sentinel-2A and 2B, all data used in this study were processed with processing baseline 2.0 or higher, data from the beginning of life until mid-2020.

The digital number is converted to be TOA reflectance as detailed in [33]. For Sentinel-2 data, the quality of pixel was determined by information provided in the MSI product quality metadata, therefore it is used to exclude cloud-contaminated pixels. The angle information was derived from metadata. For this study, the data are filtered with 2.5 sigma threshold and visual inspection.

### 2.1.4. MODIS Terra and Aqua

MODIS (Moderate-Resolution Imaging Spectroradiometer) Terra and Aqua were launched on 18 December 1999, and 4 May 2002, respectively. MODIS Terra is a morning orbit sensor crossing equatorial time at 10:30 a.m., whereas MODIS Aqua is an afternoon orbit sensor with equatorial crossing at 1:30 p.m. They acquire data in 36 spectral bands ranging from 400 nm to 2100 nm. The MODIS data products have three different spatial resolutions; 250 m, 500 m, and 1 km. with 2330 km. swath width coverage. The reported calibration uncertainties of both MODIS TOA reflectance products are approximately 2% and 1% for all spectral bands with sensor zenith angles at nadir for Terra and Aqua, respectively [34].

The digital number is converted to TOA reflectance as detailed in [35]. The angle information was derived from metadata. For this study, only Libya-4 data are used for the validation as this PICS site representing EPICS-NA spectral profile. The data are filtered with a 2.5 sigma threshold and visual inspection.

Details of all satellite data and all angles of illumination and sensor viewing geometry: satellite view zenith angle-VZA, satellite view azimuth angle-VAA, solar zenith angle-SZA and solar azimuth angle-SAA, and the number of scenes are summarized in Table 1. The location of EPICS-NA with respect to the Landsat Worldwide Reference System 2 (WRS2) paths and rows and an acquisition cycle of 16-day period for Landsat data are shown in Figure 1. Taking advantage of EPICS-NA locations across the North African region, it was clearly seen that a calibration point can be obtained nearly on a daily basis, which is far superior to utilizing a single PICS location with a revisit of every 16 days for Landsats and 10 days for Sentinel-2s.

### *2.2. The ExPAC BRDF Model*
### 2.2.1. The ExPAC Data: Hyperion EO-1, Landsat-8 Data and MODIS TERRA

Using a single PICS, existing empirical absolute models were developed using MODIS TERRA solar zenith angle ranging from 15 to 55° and Hyperion viewing angles (0° to ±25°) in a spherical degree unit to generate the BRDF model. MODIS Terra was used as a reference radiometer and for individual PICS locations [7,17]. For this study, the development of an empirical absolute model would use Landsat-8 as a reference radiometer using EPICS-NA data covering the Northern Saharan desert region. The variation of Landsat-8 viewing angles was ranging only from 3° to 8° which was very restricted to generate a BRDF model. The concept of integrating multiple satellite data to be spectrally matched with Landsat-8 was motivated. It could expand sensor viewing angles to be within ±30° and solar zenith angle up to 55 degrees using Hyperion, MODIS TERRA, and Landsat-8 data as seen in Figure 2a. The Hyperion spectral profiles over the EPICS-NA region were selected based on Shrestha et al. [36]. The data were filtered with 2.5 sigma and manual visualization. Hence, overall, 667 scenes were used in this study. All hyperspectral profiles were corrected for yearly drifts and absolute gain and bias calibration as described in [31], referred to as "Calibrated Hyperspectral profiles".

The process of converting hyperspectral data to match multispectral data was to integrate hyperspectral data with respect to the Relative Spectral Response (RSR) of that multispectral sensor and weighted by the respective RSR of the sensor at each sampling wavelength for each spectral band, as shown in Equation (1):

$$\rho_{L8\_Simulated} = \frac{\int_{\lambda_1}^{\lambda_2} \rho_\lambda \times RSR_{L8\_\lambda} d\lambda}{\int_{\lambda_1}^{\lambda_2} RSR_{L8\_\lambda} d\lambda} \tag{1}$$

where $\rho_{L8\_Simulated}$ is the Landsat-8 simulated TOA reflectance from Calibrated Hyperspectral profiles, $\rho_\lambda$ is the Hyperion Calibrated Hyperspectral profiles, and $RSR_{L8\_\lambda}$ is the Relative Spectral Response of Landsat-8 sensor.

There was some spectral discrepancy between $\rho_{L8\_Simulated}$ and Landsat-8 data. Therefore, a temporal gain was computed to bring $\rho_{L8\_Simulated}$ to be same spectral level as Landsat-8 data as the following, in Equation (2):

$$Temporal\ Gain_\lambda = \frac{\rho_{L8\_\lambda}}{\rho_{L8\_Simulated\_\lambda}} \tag{2}$$

Once the Temporal Gain was applied to $\rho_{L8\_Simulated}$, the Hyperion data were spectrally matched to Landsat-8 data.

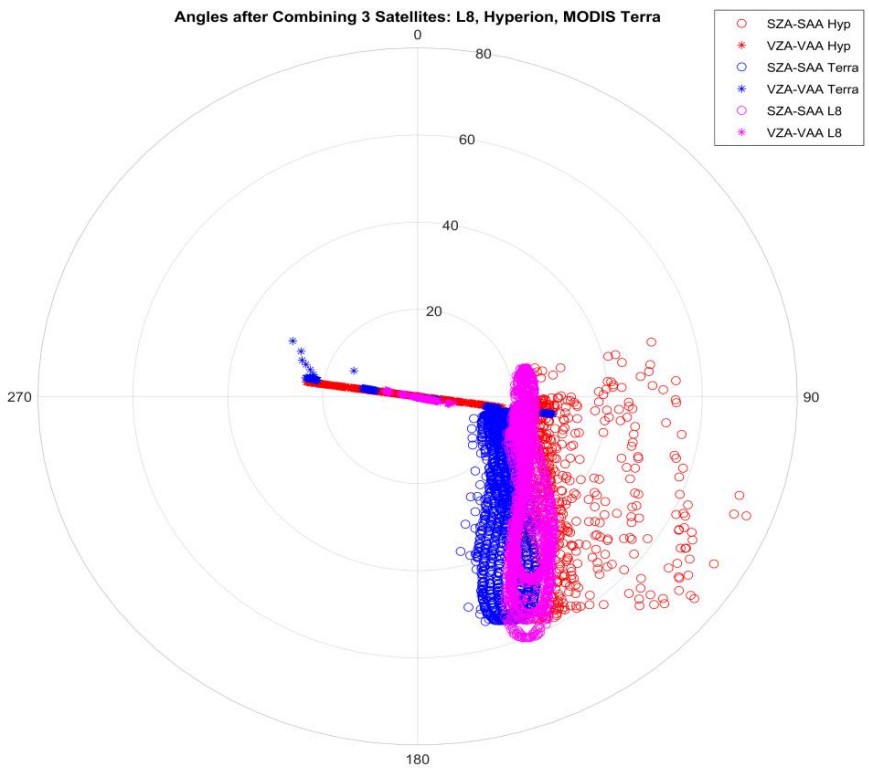

(**a**) ExPAC data with Solar and Sensor Angles.

**Figure 2.** *Cont.*

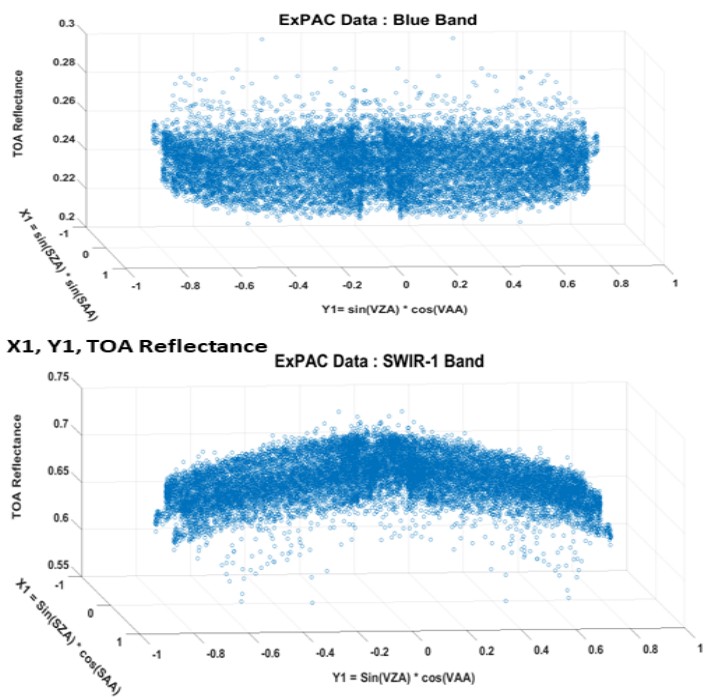

(**b**) Quadratic Relationship between TOA Reflectance and X1, Y1.

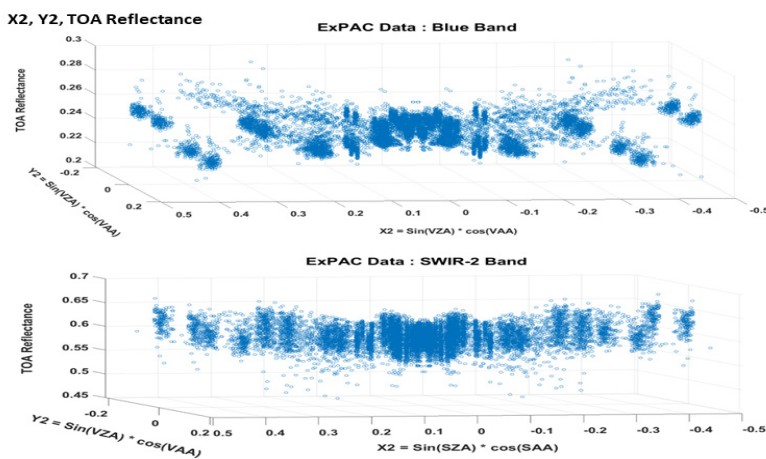

(**c**) Linear–Quadratic relationship between TOA Reflectance and X2, Y2.

**Figure 2.** (**a**) The ExPAC data with solar and viewing angles variation, (**b**) the relationship between X1, Y1 and TOA Reflectance showing a quadratic pattern, (**c**) the relationship between X2, Y2 and TOA Reflectance showing a linear–quadratic model.

Khadka et al. [37] used a scaling adjustment factor (SAF) to bring Landsat-7, Sentinel-2A, Sentinel-2B, MODIS Terra, and MODIS Aqua data to match Landsat-8. The SAF was computed by taking the mean ratio between Landsat-8 TOA reflectance and a sensor using a near co-incident acquisition with Landsat-8. Then, after applying the SAF to each sensor, the combined data from these five satellites were used to generate the 4-Angle BRDF model. Thus, for this study, the MODIS Terra Temporal Gain was computed by using Equation (2) replacing $\rho_{L8\_Simulated}$ with MODIS Terra TOA reflectance. As a result, the MODIS Terra data with Temporal Gain is forced to spectrally match with Landsat-8 and preserved its angles. The Temporal Gain results and standard deviation can be found in Table 2. Figures A2 and A3 display the comparison of MODIS Terra with Temporal Gain and Landsat-8 as a function of view zenith angles and decimal year, respectively.

**Table 2.** Temporal Gain to bring Hyperion and Terra MODIS to match Landsat-8 spectral data.

| Temporal Gain | CA | Blue | Green | Red | NIR | SWIR1 | SWIR2 |
|---|---|---|---|---|---|---|---|
| Hyp to L8 | 0.9326 ± 0.031 | 0.9766 ± 0.0274 | 0.9890 ± 0.018 | 1.0103 ± 0.030 | 0.9965 ± 0.025 | 1.0346 ± 0.032 | 1.0079 ± 0.035 |
| Terra to L8 | – | 1.0115 ± 0.002 | 1.0371 ± 0.003 | 1.0651 ± 0.006 | 1.0310 ± 0.001 | 0.9668 ± 0.002 | 0.9171 ± 0.007 |

After applying 'Temporal Gain' to MODIS Terra, the offsets between the two sensors were minimized due to the reduction in atmospheric effects, differences in the RSR, and spectral signature of the ground target [37], as seen in Figure A3.

Finally, the three-satellite data were combined into a single temporal dataset that spectrally matched Landsat-8 and preserved angles from images, which is referred to as 'ExPAC data'. The ExPAC data show agreement better than 2% for all spectral bands, as seen in Table 3.

**Table 3.** Mean TOA Reflectance of Hyperion to Landsat-8 and Terra MODIS to match Landsat-8 data after applied Temporal Gain to create ExPAC data.

| ExPAC Data | CA | Blue | Green | Red | NIR | SWIR1 | SWIR2 |
|---|---|---|---|---|---|---|---|
| Hyp-L8 | 0.2283 | 0.2447 | 0.3396 | 0.4733 | 0.5896 | 0.6770 | 0.5883 |
| Terra-L8 | NA | 0.2441 | 0.3352 | 0.4877 | 0.6066 | 0.6784 | 0.5724 |
| Landsat-8 | 0.2281 | 0.2445 | 0.3400 | 0.4736 | 0.5904 | 0.6814 | 0.5938 |
| Mean | 0.2282 | 0.2444 | 0.3382 | 0.4782 | 0.5956 | 0.6789 | 0.5848 |
| Std. Dev | 0.0001 | 0.0003 | 0.0027 | 0.0082 | 0.0096 | 0.0023 | 0.0112 |
| CV (%) | 0.06% | 0.13% | 0.79% | 1.72% | 1.61% | 0.34% | 1.91% |

### 2.2.2. The Generation of the 4-Angle BRDF Model

Based on physical properties of surface reflectance, TOA reflectance of a target is varied due to different illumination and viewing angles which can be described by Bidirectional Reflectance Distribution Function (BRDF) model. Prior to Landsat Collection-1 data, when considering BRDF model for radiometric analysis, many research works used a simple empirical BRDF model using a linear trend of TOA reflectance as a function of solar zenith angle and viewing zenith angle. These angles were extracted from metadata files and solely specified at the scene-center angle. Using data from Collection-1 and -2, all information on the Landsat angles: sensor viewing geometry and solar illumination geometry, which referred to as the four angles. These angles can be retrieved per pixel via the angle bands.

Instead of using all four angles in spherical degree units to develop a BRDF Model, this research adopted angles in Cartesian coordinates as described in [24] and the Equations to perform angle conversion are as follows:

$$X_1 = \sin(\theta_{SZ}) * \sin(\theta_{SA}) \tag{3}$$

$$Y_1 = \sin(\theta_{SZ}) * \cos(\theta_{SA}) \tag{4}$$

$$X_2 = \sin(\varphi_{VZ}) * \sin(\varphi_{VA}) \tag{5}$$

$$Y_2 = \sin(\varphi_{VZ}) * \cos(\varphi_{VA}) \tag{6}$$

where $\theta_{SZ}$ = Solar Zenith Angle, $\theta_{SA}$ = Solar Azimuth Angle, $\varphi_{VZ}$ = Satellite Viewing Angle, $\varphi_{VA}$ = Satellite Azimuth Angle.

It can be clearly seen that the relationship between TOA reflectance and the solar illumination geometry is quadratic, especially in longer wavelengths when ExPAC data were plotted against $X_1$ *and* $Y_1$. The dataset was repeated and mirrored into four quadrants

as seen in Figure 2b, whereas the relationship with $X_2,Y_2$ was a linear relationship as shown in Figure 2c. Farhad et al. has suggested using the 4-Angle BRDF with quadratic fit and multi-linear fit with all interaction terms to represent the BRDF of sand targets [24].

$$Reflectance = \beta_0 \quad +\beta_1 X_1^2 + \beta_2 Y_1^2 + \beta_3 X_2^2 + \beta_4 Y_2^2 + \beta_5 X_1 Y_1 + \beta_6 Y_1 Y_2 + \beta_7 X_2 Y_2 + \beta_8 X_2 Y_1 \\ +\beta_9 Y_1 Y_2 + \beta_{10} X_1 X_2 + \beta_{11} X_1 + \beta_{12} Y_1 + \beta_{13} X_2 + \beta_{14} Y_2 \tag{7}$$

where $\beta_0$, $\beta_1$, $\beta_2$, $\beta_3$, $\beta_4$ ... $\beta_{14}$ are the coefficients of the model which are calculated with known parameters $Y_1$, $X_1$, $Y_2$, $X_2$ and Reflectance for quadratic, linear and all interaction terms.

In order to simplify the 4-Angle BRDF model generated from Equation (7) for an empirical BRDF model for absolute calibration, Das Chaity et al. tested each BRDF model parameter for significant contribution using Student's $t$-test analysis and selected only the top two parameters that were statistically significant to develop the absolute calibration model. By assuming no interactions between these four angles and considering the quadratic terms of $X_1^2,Y_1^2$ and linear terms of $X_2,Y_2$ [10]. Thus, this study also used similar criteria and tested to select significant parameters that are statistically significant to all seven spectral bands. The selection was done and considered all interaction parameters. It was found that the impact of $X_1,Y_1,X_2,Y_2,X_1Y_1, X_1Y_2,X_2Y_2,X_1Y_2$ parameters were not statistically significant, especially in longer wavelengths. Table A2 displays the analysis of Student's $t$-test and the hypothesis test results for the SWIR-1 band. Hence, in order to simplify the 4-Angle BRDF model for the development of the Extended PICS Absolute Calibration model, only six BRDF parameters and BRDF–intercept are considered—Equation (8) as follows:

$$\rho_{Model} = \beta_0 + \beta_1 X_1^2 + \beta_2 Y_1^2 + \beta_3 X_2^2 + \beta_4 Y_2^2 + \beta_5 X_1 X_2 + \beta_6 Y_1 Y_2 \tag{8}$$

The results of all six coefficient parameters: $X_1^2,Y_1^2,X_2^2,Y_2^2,X_1X_2, Y_1Y_2$ and intercepts are summarized in Table 4. Figure 3 displays the histogram of ratios between satellite measurements and model-predicted values between the selected seven parameters 4-Angle BRDF model in Equation (8) versus the fifteen parameters 4-Angle BRDF model in Equation (7). They are mostly on top of each other, giving similar distribution. Thus, these seven selected parameters are the best combination for the development of an empirical absolute calibration model. These coefficients from the 4-Angle BRDF model were then plotted as a function of Landsat-8 center wavelengths for each parameter, and additional curves were fitted to the data, i.e., creating hyperspectral BRDF coefficients model from Landsat-8 based ExPAC data multispectral BRDF coefficients [9,21]. These polynomial fits represent hyperspectral 4-Angle BRDF coefficients model in the ExPAC model for $\beta_1$, $\beta_2$, ... , $\beta_6$ as in Equation (7). Figure 4a–f show curve fitting models for all six BRDF parameters; the low RMSE and high adjusted R-Squared value indicate the model fit data are well within 1 sigma accuracy.

**Table 4.** The ExPAC Model Coefficients for specific wavelengths.

| Center Wavelength (nm) | CA (440) | Blue (480) | Green (545) | Red (655) | NIR (865) | SWIR-1 (1610) | SWIR-2 (2200) |
|---|---|---|---|---|---|---|---|
| $X_1^2$ | 0.0234 | 0.0511 | 0.0162 | 0.0530 | 0.0253 | −0.0588 | −0.0724 |
| $Y_1^2$ | 0.0098 | 0.0051 | −0.0091 | −0.0255 | −0.0386 | −0.0727 | −0.0552 |
| $X_2^2$ | 0.1396 | 0.0699 | 0.0715 | 0.0280 | 0.0108 | 0.0722 | 0.1289 |
| $Y_2^2$ | −1.3725 | −1.5704 | −1.9356 | −2.3844 | −2.1897 | −1.9655 | −2.0677 |
| $X_1X_2$ | −0.0014 | −0.0042 | 0.0113 | 0.0024 | 0.0156 | 0.0094 | 0.0063 |
| $Y_1Y_2$ | 0.2828 | 0.2949 | 0.1510 | 0.0894 | 0.1375 | 0.1754 | 0.1331 |
| Intercept | 0.2235 | 0.2335 | 0.3381 | 0.4671 | 0.5890 | 0.7005 | 0.6141 |

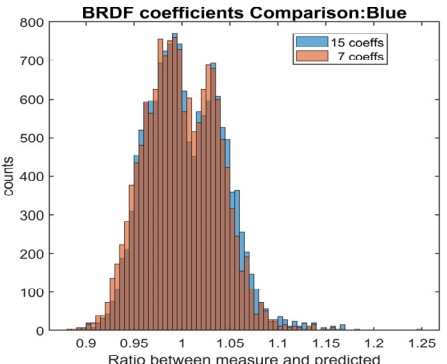
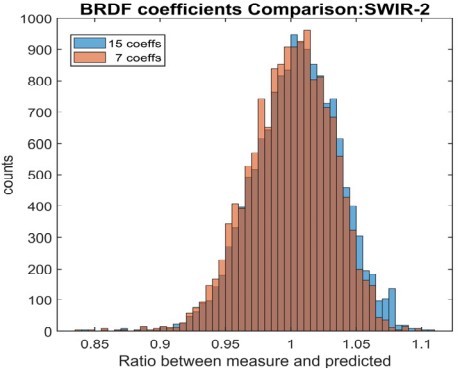

**Figure 3.** The histogram of ratios between satellite measurements and model predicted the comparison of the selected 7 parameters 4-Angle BRDF model-Equation (8) versus the 15 parameters 4-Angle BRDF model-Equation (7) for Blue and SWIR-2 band, which are mostly on top of each other.

### 2.2.3. The Adjustment Factor for ExPAC Model

As described in Helder et al. and Mishra et al., using Terra MODIS as the radiometer reference and Hyperion hyperspectral as the spectral profile, the scale factor was derived to scale Hyperion spectrum so that, when integrated over the MODIS spectral bandpass, it will produce the comparable TOA reflectance of Terra MODIS [21,38]. For this study, Landsat-8 OLI is the radiometer reference and its sensor calibration is well within 3% [39]. The adjustment factor was in fact a Pseudo Cross-Cal Gain (*XCal Gain*) that will place the sensor's derived TOA reflectance to match ExPAC data BRDF Intercept, as described in Section 2.2.2 and shown in Figure 4g. The *XCal Gain* will be used to normalize the derived TOA reflectance from the ExPAC hyperspectral profile to match any sensor, as described below:

$$\rho_{L8}(\lambda) = \frac{\int_{\lambda_1}^{\lambda_2} \rho_{Hyp} RSR_{L8} d\lambda}{\int_{\lambda_1}^{\lambda_2} RSR_{L8} d\lambda} \qquad (9)$$

$$XCal\ Gain(\lambda) = \frac{\beta_0(\lambda)}{\rho_{L8}(\lambda)} \qquad (10)$$

where $\rho_{Hyp}$ is EPICS-NA hyperspectral profiles, $RSR_{L8}$ is Landsat-8 Relative Spectral Response, $\rho_{L8}$ is the Landsat-8 derived TOA reflectance from 58 EPICS-NA hyperspectral profiles, $\beta_0$ is the Landsat BRDF-Intercept derived from the ExPAC Model.

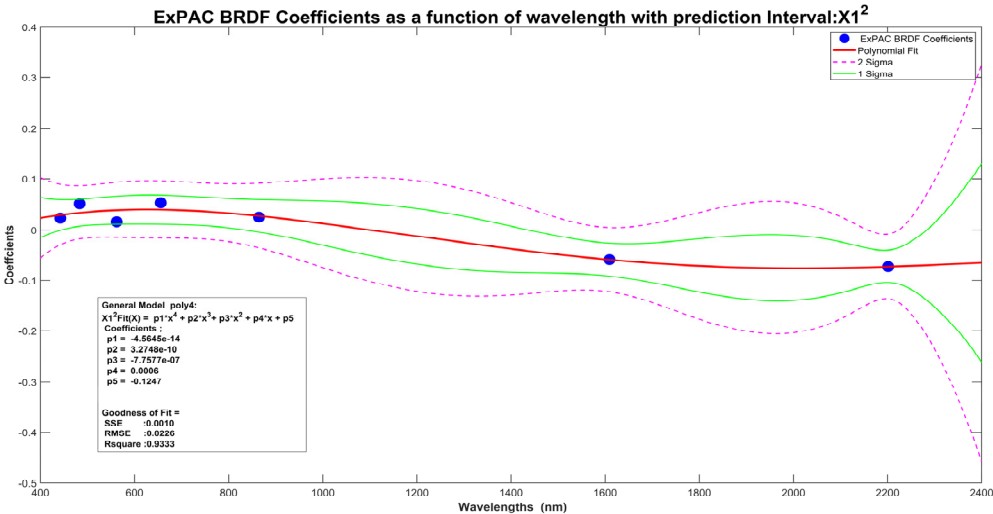

(**a**) $X_1^2$ ($\beta_1$) coefficients and a polynomial fit.

**Figure 4.** *Cont.*

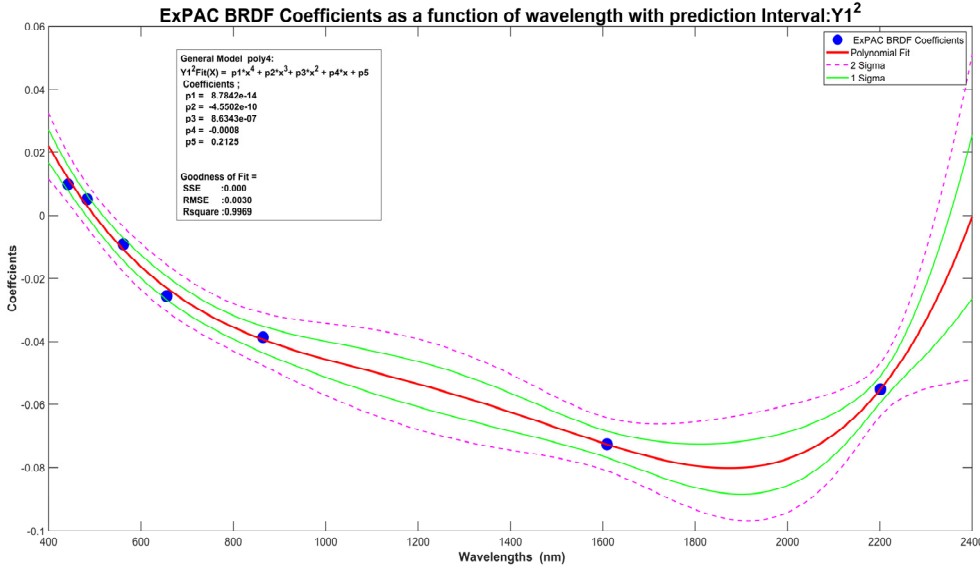

(**b**) $Y_1^2$ ($\beta_2$) coefficients and a polynomial fit.

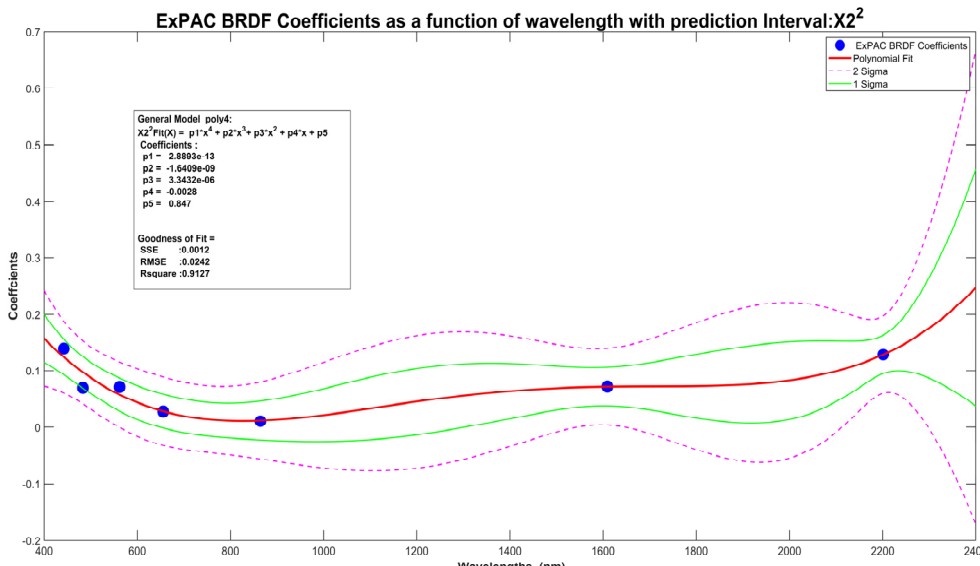

(**c**) $X_2^2$ ($\beta_3$) coefficients and a polynomial fit.

(**d**) $Y_2^2$ ($\beta_4$) coefficients and a polynomial fit.

**Figure 4.** *Cont.*

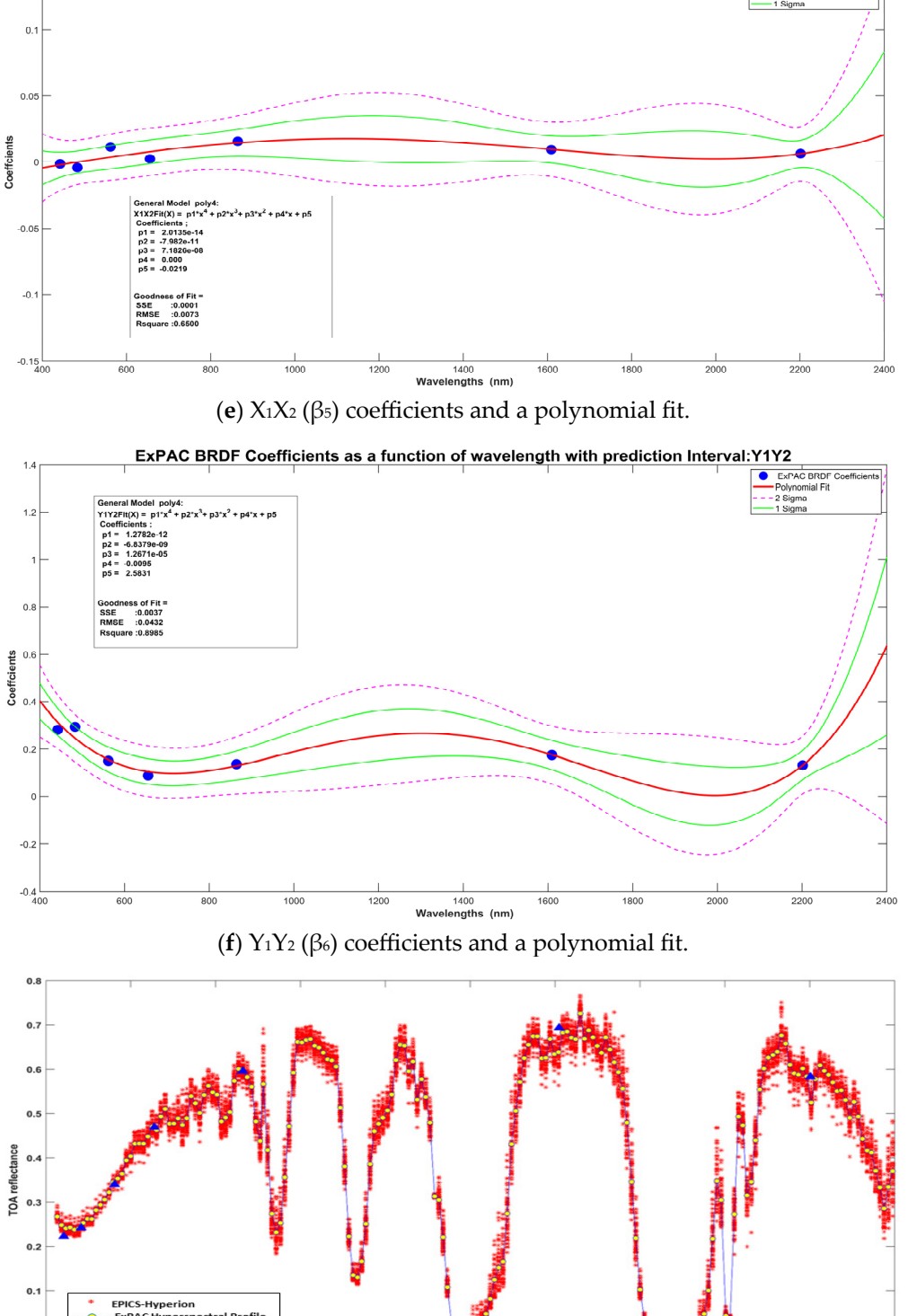

(**e**) $X_1X_2$ ($\beta_5$) coefficients and a polynomial fit.

(**f**) $Y_1Y_2$ ($\beta_6$) coefficients and a polynomial fit.

(**g**) ExPAC Hyperspectral profile shown in yellow, 7 ExPAC Intercepts ($\beta$0) are shown in blue triangles, Hyperion data-58 scenes shown in red.

**Figure 4.** (**a**–**f**) The ExPAC 4-Angle BRDF Model fits for $X_1{}^2$, $Y_1{}^2$, $X_2{}^2$, $Y_2{}^2$, $X_1X_2$, $Y_1Y_2$ coefficients showing details of polynomial fits; (**g**) shows the ExPAC hyperspectral profile and BRDF-Intercept.

EPICS-NA hyperspectral profiles were selected using 6 days near co-incident pairs Hyperion–Landsat-8, with a viewing angle less than $\pm 5°$, and a solar zenith angle of $30 \pm 5°$; total of 58 scenes were chosen, as shown in red, Figure 4g. The $\rho_{Hyp}$, denoted as ExPAC hyperspectral profiles of all 58 EPICS-NA hyperspectral profiles as shown in red Figure 4g. The $\rho_{L8}$ is a derived TOA Reflectance, computed by a ratio between the integration of Landsat-8's RSR (relative spectral response), with EPICS-NA hyperspectral profiles weighted by its RSR at a sampling wavelength, Equation (9). The *XCal Gain* was then computed by a ratio between BRDF-Intercept $\beta_0(\lambda)$ and derived TOA Reflectance ($\rho_{L8}$)—Equation (10). The final *XCal Gain* values were obtained by averaging the computed 58-*XCal Gain*$(\lambda)$ as shown in Table 4. The average *XCal Gain* values agree to be within 3.5% for all bands. The *XCal Gain* values were organized to cover range of Hyperion spectral bands at 1 nm resolution as follows: using stepwise function for wavelength corresponding to Coastal Aerosol band (400–440 nm), blue band (440–475 nm), SWIR-1 (960–1850 nm), and SWIR-2 (1850–2395 nm). Wavelength corresponding to green (475–590 nm), red (590–830 nm), and NIR (835–960 nm), the *XCal Gain* values were computed using linear interpolation, a similar procedure to that described by Kaewmanee et al. [22].

### 2.2.4. The Extended PICS Absolute Calibration Model–ExPAC Model

The development of the BRDF coefficients fits were simply empirically fit to allow the prediction of BRDF coefficients at a given wavelength with respect to $X_1{}^2$, $Y_1{}^2$, $X_2{}^2$, $Y_2{}^2$, $X_1X_2$, $Y_1Y_2$ coefficients derived from ExPAC data. A similar approach was performed to develop PICS Absolute Calibration model (APICS), generating hyperspectral BRDF coefficients model from multispectral BRDF coefficients [9,21]. For each parameter, the BRDF coefficients were plotted as a function of Landsat-8 Center wavelengths; then, a curve was fitted with a 4th order polynomial as shown in Figure 4.

The simplified model for the Extended PICS Absolute Calibration Model is formed as:

$$\rho_{Model}(\lambda, X_1, Y_1, X_2, Y_2) = \rho_h(\lambda) * XCal_{Gain}(\lambda) + X_1^2 \cdot \beta_1(\lambda) + Y_1^2 \cdot \beta_2(\lambda) + X_2^2 \cdot \beta_3(\lambda) + Y_2^2 \cdot \beta_4(\lambda) + X_1 X_2 \cdot \beta_5(\lambda) + Y_1 Y_2 \cdot \beta_6(\lambda) \quad (11)$$

where $XCal_{Gain}(\lambda)$ represents the Pseudo Cross Cal gain to place BRDF intercept matching any sensor as described in Section 2.2.3, $\rho_h(\lambda)$ is the ExPAC hyperspectral profile which is the average of all 58 EPICS-NA hyperspectral profiles as shown in yellow Figure 4g, $\beta_1(\lambda)$, $\beta_2(\lambda)$, $\beta_3(\lambda) \dots \beta_6(\lambda)$ represent the hyperspectral BRDF coefficients model for $X_1^2$, $Y_1^2$, $X_2^2$, $Y_2^2$, $X_1 X_2$ and $Y_1 Y_2$ as described in Section 2.2.2, as summarized in Table 5. Kaewmanee et al. and Barsi et al. had stated that a correction for atmosphere was very small and could be negligible for an absolute calibration model over Libya-4; therefore, the ExPAC model does not include atmospheric parameters [21,30].

**Table 5.** XCal Gain values derived from 58 EPICS-NA hyperspectral profiles with the average values and standard deviation.

|  | CA | Blue | Green | Red | NIR | SWIR-1 | SWIR2 |
|---|---|---|---|---|---|---|---|
| *XCal Gain*$(\lambda)$ | 0.9148 | 0.9334 | 0.9837 | 0.9972 | 0.9946 | 1.0645 | 1.0435 |
| Std. Dev. | 0.030 | 0.026 | 0.017 | 0.030 | 0.024 | 0.032 | 0.036 |
| CV(%) | 3.27% | 2.81% | 1.77% | 3.01% | 2.46% | 3.04% | 3.44% |

The performance of ExPAC model is quantified in terms of accuracy, precision, systematic bias and relative accuracy. The output of the model gives four matrices: firstly, accuracy ($A_{Accuracy}$) shows the average bias between the model prediction and satellite measurements—Equation (12). Secondly, the precision ($P_{Precision}$) of the model shows the repeatability of the estimates, the RMSE of model prediction and satellite measurements with respect to $A_{Accuracy}$—Equation (13). Thirdly, the systematic offset or bias as root-mean-square error (RMSE) represents the standard deviation of model residuals, as shown in Equation (14). It is noted that the $A_{Accuracy}$, RMSE and $P_{Precision}$ matrices are computed in

reflectance unit [28]. The fourth matric is a percentage of relative accuracy between the measure and estimates as in Equation (15).

$$A_{Accuracy} = \frac{1}{N}\sum_{i=1}^{N}\left(\rho_{Model\_i} - \rho_{Measure\_i}\right) \tag{12}$$

$$P_{Precision} = \sqrt{\frac{1}{N-1}\sum_{i=1}^{N}\left(\rho_{Model_i} - \rho_{Measure_i} - A_{Accuracy}\right)^2} \tag{13}$$

$$RMSE = \sqrt{\frac{1}{N}\sum_{i=1}^{N}\left(\rho_{Model_i} - \rho_{Measure_i}\right)^2} \tag{14}$$

$$Model\ Accuracy = \frac{A_{Accuracy}}{\frac{1}{N}\sum_{i=1}^{N}\left(\rho_{Measurel\_i}\right)} \times 100\% \tag{15}$$

where $\rho_{Model\_i}$ *and* $\rho_{Measure\_i}$ are model predicted and satellite measurement TOA reflectance, respectively, N is the total number of satellite measurements.

## 3. EPICS-Based Cross-Calibration

Shrestha et al., 2019 [1], described the potential of using EPICS-NA for cross-calibration and compared results with PICS-based cross-calibration using Landsat-8 and Sentinel-2A. The results showed that these two methods gave similar results to within 2% for all spectral bands except SWIR bands with ~4%. Recently, Fajardo et al. improved the capability of finding stable pixels to a global level (EPICS-Global Classification), modifying the criteria of the K-mean algorithm and no threshold requirements using Landsat-8 data 30 m spatial resolution. The details of finding EPICS-Global Classification which resulted in 160 classifications can be found in [4]. Recent modification of the EPICS-Global Classification aimed to isolate mixed spectral features into 300 classifications [20]. During Landsat-9 OIV, the inter-comparison between Landsat-8 and Landsat-9 used a similar methodology to monitor the performance of Landsat-9 in orbit on a weekly basis. This section presents the performance of EPICS-based cross-calibration with EPICS-NA and EPICS-Global. Then, the methods would be expanded to perform inter-comparison of Landsat satellites and Sentinel-2 satellites.

### 3.1. Spectral Band Adjustment Function (SBAF)

Both Landsat-8 and Landsat-9 were designed to possess identical spectral bands properties. However, there are some differences, expected to be very small, in spectral bandpasses which can be calculated. Details of SBAF calculation can be found in [1,40]. For this study, there were 667 Hyperion scenes over EPICS-NA for SBAF calculation. The SBAF values are summarized in Table 6.

**Table 6.** SBAF values to bring Landsat-9 matching Landsat-8.

|  | CA | Blue | Green | Red | NIR | SWIR-1 | SWIR-2 |
|---|---|---|---|---|---|---|---|
| SBAF | 1.000 | 1.000 | 1.005 | 1.002 | 1.000 | 1.000 | 1.001 |
| Std. Dev. | 0.000 | 0.000 | 0.001 | 0.000 | 0.000 | 0.000 | 0.000 |

### 3.2. Bidirectional Reflection Distribution Function (BRDF)

For the purpose of inter-comparison between Landsat-8 and Landsat-9, the 4-Angle BRDF model was generated using EPICS-NA and EPICS-Global Landsat-8 Collection-2 data separately. Landsat-9 data would be spectrally matched with Landsat-8 by applying the SBAF values as shown in Table 6. Details of generating the 4-Angle BRDF model were explained in Section 2.2.2. The generated 4-Angle BRDF model for each EPICS target would be used for BRDF normalization for both satellites. The reference angles were selected to perform BRDF normalization for both satellites; the normalized TOA reflectance is

computed as in Equation (16): the reference TOA reflectance is multiplied by the ratio of the measure and model predicted.

$$\rho_{\text{normalized}} = \frac{\rho_{measure}}{\rho_{model}} * \rho_{Ref} \tag{16}$$

where $\rho_{normalized}$ is the normalized TOA reflectance, $\rho_{Ref}$ is the predicted TOA reflectance derived from 4-Angle BRDF model with reference angles, $\rho_{measure}$ is the estimated TOA reflectance from the image, $\rho_{model}$ is the predicted TOA reflectance derived from the 4-Angle BRDF model with angles from the image.

### 3.3. The EPICS-Based Cross-Calibration Ratio

The EPICS-based cross-calibration ratio, for both EPICS-NA and EPICS-Global, was computed using 7 days near co-incident pairs data. The ratios between normalized TOA reflectance Landsat-8 and Landsat-9 were calculated and the average of all derived ratios represents the difference between Landsat-8 and Landsat-9, using data from November 2021 to January 2022.

### 4. The ExPAC Double Ratio and the EPICS-Based Cross-Calibration Uncertainties Analysis

### 4.1. The ExPAC Double Ratio Uncertainties Analysis

For the purpose of the OIV test, the main objective was to determine how Landsat-9 performs in orbit with respect to Landsat-8. The inter-comparison between these two satellites using the method 'ExPAC Double Ratio' was used to determine the differences on a weekly basis. The uncertainties of the ExPAC Double Ratio were studied to ensure that the differences or the ratio were within an acceptable level of uncertainty. The study was designed using Sentinel-2A and Sentinel-2B on a randomly selected day, 3 months after the Sentinel-2B launch date; 23 September 2015 was selected as day 1. Then, using Monte Carlo Simulation for each iteration, day 1 was randomly selected to start; the ExPAC Ratio was then generated from wk1 to wk25 for both satellites. The ExPAC Double Ratio (D-Ratio) was then calculated, depicted as MeanRatio from wk1 to wk25. The process repeats for 1000 randomly selected start days. Finally, the MeanRatio values from wk1 with 1000 data points to wk25 with 1000 data points were generated. Then, the standard deviation of MeanRatio for each week was calculated, k = 1. Figure 5 displays the flowchart of ExPAC Double Ratio uncertainties.

The ExPAC Double Ratio Uncertainties were determined using 3-sigma (k = 3) ensuring that 99.7% of data was included. Figure 5 shows a flow diagram of the determination of ExPAC Double Ratio Uncertainties analysis.

The 3-sigma ExPAC Double Ratio uncertainties were displayed in Figure 6a. It can be clearly seen that using only 1 week of data, the uncertainties were within 6% for all bands. As more data was gathered weekly, the convergence of uncertainties was found to be at 2% level after 20 weeks. The reported ExPAC Double Ratio uncertainties were based on 25-week data and would be used for inter-comparison analysis results as shown in a table attached to Figure 6a.

### 4.2. The EPICS-Based Cross-Calibration Ratio Uncertainties Analysis

For EPICS-based cross-calibration ratio uncertainties analysis, the same process as described in Section 4.1 was carried out for both targets using the 'EPICS Cross Cal ratio' between Sentinel-2A and Sentinel-2B. As more data were gathered weekly, the convergence of uncertainties was found to be better than 2.5% level for all spectral bands after 20 weeks for both EPICS-NA and EPICS-Global as shown in Figure 6b. The EPICS-based cross-calibration ratio uncertainties were determined using 3-sigma (k = 3) ensuring that 99.7% of data were included.

# Finding ExPAC Double Ratio: Uncertainty

**Figure 5.** The flowchart of the ExPAC Double Ratio Uncertainties Analysis.

The reported EPICS-based cross-calibration uncertainties were based on 25-week data and would be used for inter-comparison analysis results, as shown in Table 7.

**Table 7.** Model Accuracy (%), RMSE, and Precision in reflectance units between the ExPAC Model predictions and all EPICS-NA measurements for 6 satellites. The Wilcoxon Rank Sum Test results are included showing that the two populations have the same median (h = 0).

| | RMSE (Reflectance Units) | | | | | | Precison (Reflectance Units) | | | | | |
|---|---|---|---|---|---|---|---|---|---|---|---|---|
| | Terra | Aqua | L7 | L8 | S2A | S2B | Terra | Aqua | L7 | L8 | S2A | S2B |
| CA | – | – | – | 0.012 | 0.021 | 0.021 | – | – | – | 0.013 | 0.020 | 0.020 |
| Blue | 0.007 | 0.011 | 0.027 | 0.014 | 0.016 | 0.016 | 0.007 | 0.010 | 0.018 | 0.017 | 0.015 | 0.014 |
| Green | 0.015 | 0.011 | 0.028 | 0.013 | 0.009 | 0.011 | 0.009 | 0.010 | 0.018 | 0.020 | 0.008 | 0.009 |
| Red | 0.029 | 0.013 | 0.041 | 0.014 | 0.017 | 0.011 | 0.016 | 0.011 | 0.027 | 0.020 | 0.013 | 0.011 |
| NIR | 0.029 | 0.016 | 0.024 | 0.012 | 0.025 | 0.013 | 0.016 | 0.014 | 0.021 | 0.019 | 0.016 | 0.011 |
| SWIR-1 | 0.021 | – | 0.020 | 0.016 | 0.029 | 0.034 | 0.015 | – | 0.017 | 0.019 | 0.021 | 0.024 |
| SWIR-2 | 0.023 | 0.022 | 0.027 | 0.019 | 0.024 | 0.024 | 0.022 | 0.021 | 0.022 | 0.020 | 0.021 | 0.021 |

| | Model Accuracy (%) | | | | | | The Wilcoxon Rank Sum Test (h) | | | | | |
|---|---|---|---|---|---|---|---|---|---|---|---|---|
| | Terra | Aqua | L7 | L8 | S2A | S2B | Terra | Aqua | L7 | L8 | S2A | S2B |
| CA | – | – | – | −0.75 | 0.43 | 1.85 | – | – | – | 0 | 0 | 0 |
| Blue | −0.63 | 0.65 | 3.51 | −2.22 | −0.20 | 0.50 | 0 | 0 | 0 | 0 | 0 | 0 |
| Green | 2.12 | 0.89 | 2.25 | −2.41 | −1.56 | −1.04 | 0 | 0 | 0 | 0 | 0 | 0 |
| Red | 3.20 | 0.88 | 2.11 | −1.71 | −1.47 | −1.39 | 0 | 0 | 0 | 0 | 0 | 0 |
| NIR | 2.45 | 0.89 | 0.57 | −1.38 | −1.23 | −0.78 | 0 | 0 | 0 | 0 | 0 | 0 |
| SWIR-1 | 1.17 | – | 1.48 | −0.93 | 1.25 | −0.22 | 0 | – | 0 | 0 | 0 | 0 |
| SWIR-2 | −0.52 | 0.53 | −0.94 | −0.42 | −1.23 | 1.42 | 0 | 0 | 0 | 0 | 0 | 0 |

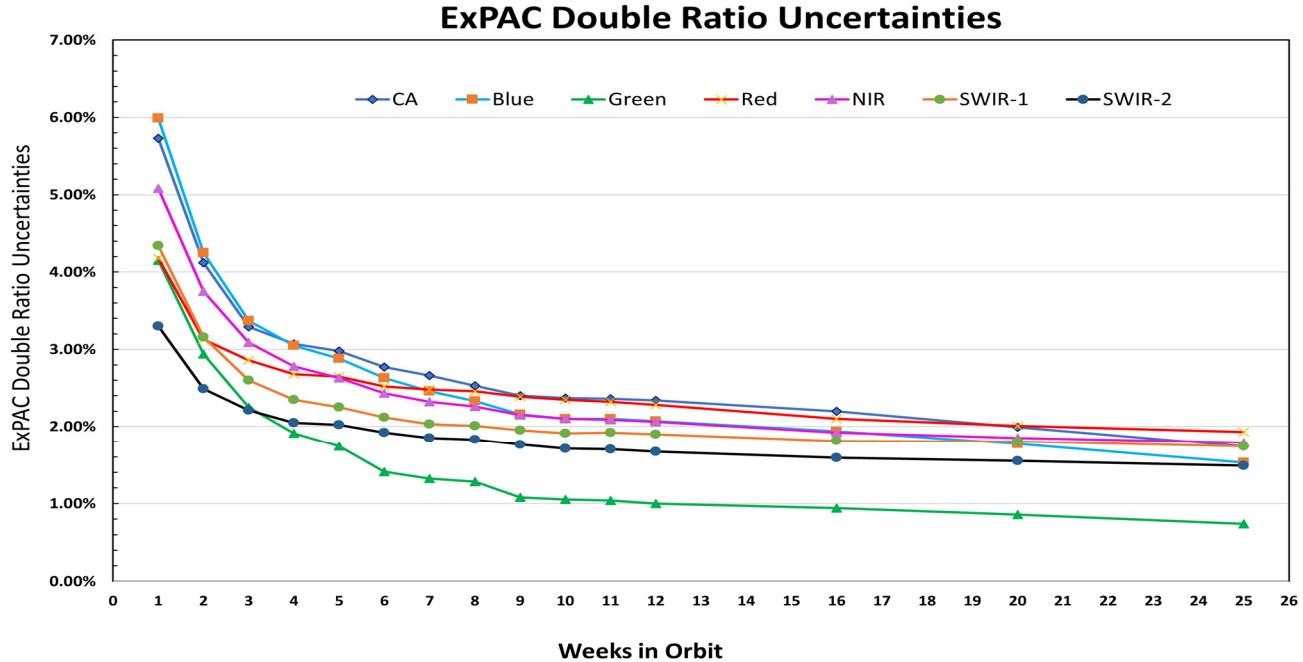

(**a**) The ExPAC Double Ratio Uncertainties.

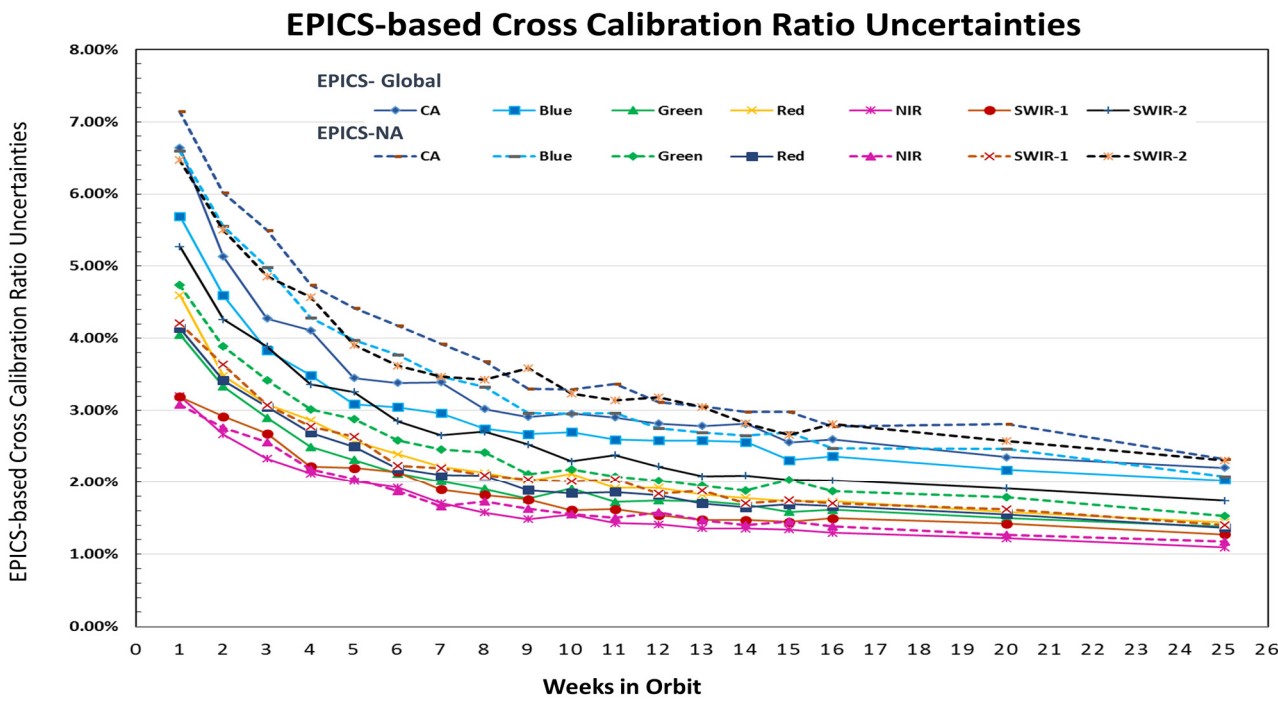

(**b**) The EPICS-based Cross-Calibration Ratio Uncertainties: EPICS-NA and EPICS-Global.

**Figure 6.** (**a**) The ExPAC Double Ratio Uncertainties; (**b**) the EPICS-based Cross-Calibration Ratio Uncertainties.

## 5. Results and Discussion

This section presents the validation of the ExPAC Model and its application. The EPICS-based cross-calibration results between Landsat missions and Sentinel2 missions will be included. The application of these three advanced methods during the Landsat-9 on-orbit initialization and verification (OIV) test will also be discussed in this section.

### 5.1. Validation of the ExPAC Model

5.1.1. The Validation of ExPAC Model with Landsat-8 Collection-2 Data

The ExPAC model was generated using Hyperion, Landsat-8, and Terra MODIS data, with the harmonized data using EPICS-NA as a target. The parameters of the model consist of hyperspectral profile of the EPICS-NA, hyperspectral coefficients for $X_1^2$, $Y_1^2$, $X_2^2$, $Y_2^2$, $X_1 X_2$, $Y_1 Y_2$ and XCal Gain as in Equation (11).

Figure 7 demonstrates the performance of the ExPAC model to Landsat-8 Collection-2 data in all spectral bands. The boxplot diagram shows the median values between the satellite measurement and the model predicted. A non-parametric test 'The Wilcoxon Rank Sum Test' was performed to test if the medians of these two datasets are having the same median. It can be concluded that the ExPAC model predicted TOA reflectance for all Landsat-8 spectral bands having the same median as the satellite measurements; the $p$-value is greater than 0.05 and h = 0, as detailed in a table in Figure 7. The overall accuracy matrices: Model Accuracy, RMSE, and Precision, were shown in Figure 8 and Table 7.

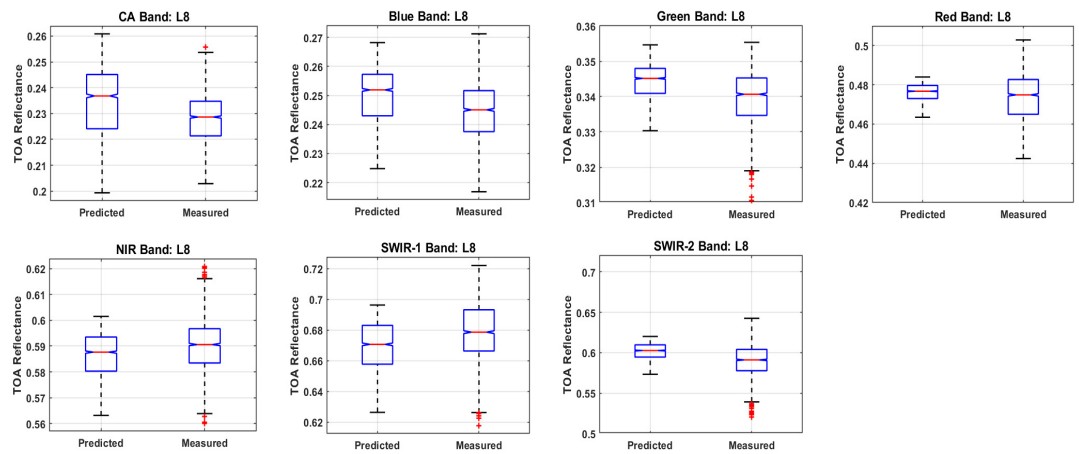

| Bands | CA | Blue | Green | Red | NIR | SWIR-1 | SWIR-2 |
|---|---|---|---|---|---|---|---|
| $p$-value | 0.0865 | 0.3734 | 0.3166 | 0.6732 | 0.6157 | 0.5939 | 0.2165 |
| h | 0 | 0 | 0 | 0 | 0 | 0 | 0 |

**Figure 7.** The performance of the ExPAC model with Landsat-8 validation: showing the boxplot of satellite measurements (measured) and model predicted (predicted) TOA reflectance. The red line displays within each box indicating the median value of these two datasets. The non-parametric test 'The Wilcoxon Rank Sum Test' results show that if h = 0, accepting the null hypothesis, then the medians of the two populations are the same.

5.1.2. The Validation of ExPAC Model Using Satellite Measurements

The satellite measurements to validate the ExPAC model are from well-calibrated satellites as mentioned earlier namely Landsat-7 ETM+, Sentinel-2A MSI, and Sentinel-2B MSI which the stated calibration uncertainties are 5%, 3–5%, and 3–5%, respectively [15] [41,42]. It should be noted that the Landsat-8 OLI and Landsat-7 ETM+ with Collection-2 data were cross-calibrated [43], whereas Sentinel-2A and Sentinel-2B were calibrated independently of Landsat-8 OLI. These four satellites are nadir-looking sensors. Terra MODIS and Aqua MODIS data were used to test whether the ExPAC Model can accommodate large solar illumination and sensor viewing angle variation. The total number of images for each satellite was summarized in Table 1.

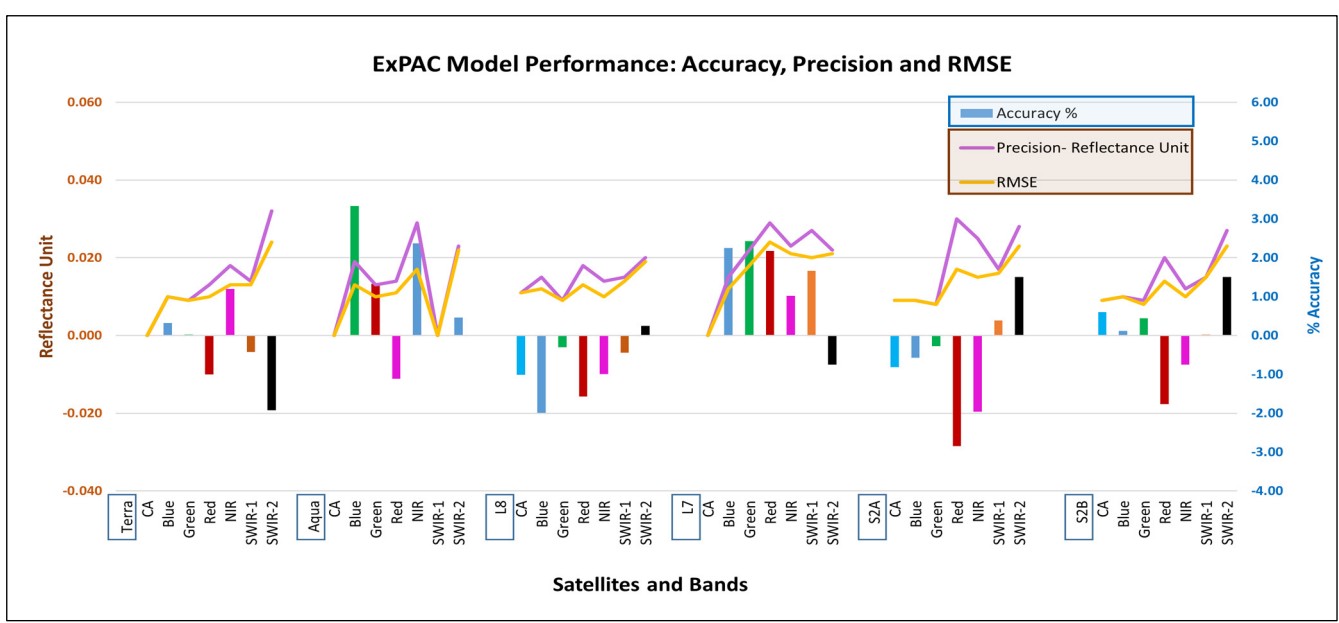

**Figure 8.** Statistical matrices of ExPAC Model performance performed on satellites, showing Percentage of Model Accuracy (Blue-axis), Precision and RMSE are shown in the reflectance unit (Brown-axis).

It can be clearly seen that all six satellites possess a similar level of systematic offsets and random error of up to 0.02 (2 reflectance units) in Coastal Aerosol and blue band and better than three reflectance units for all other bands, details in Table 7 and Figure 8. The Wilcoxon Rank Sum Test also indicates that the satellite measurements and model predicted are having same medians (h = 0) for all six satellites, details in Table A1. The repeatability of prediction (Precision) is well within two reflectance units (0.02) for all satellites and all bands. The accuracy of the model gives an overall performance at 3% level for all spectral bands except Landsat-7 at ~2–3.5% in visible bands and 1.5% for short-wave infrared bands. The ExPAC model can also accommodate large field of view (FOV) satellites: Terra MODIS and Aqua MODIS (FOV of ±49.5°), with 1–3% accuracy for all six spectral bands, as shown in Table 7.

5.1.3. The ExPAC Model vs. Non-Landsat-8 Equivalent Spectral Bands

The ExPAC Model was validated using satellites with spectral bands equivalent to Landsat-8 in visible and short-wave infrared bands. The performance was well within 3% accuracy, as shown in Figure 8. Das Chaity et al. tested the hyperspectral empirical absolute calibration model with non-Landsat-8 bands using Sentinel-2A and 2B and found that the model did not perform very well, giving accuracy within 3.75% and 3% for Sentinel-2A and 2B, respectively, due to the lack of information to determine cross-scale factors in the Red Edge and broadband NIR bands [10]. Therefore, the ExPAC model should also be tested to identify the performance against non-Landsat-8 equivalent spectral bands using Sentinel-2A and 2B in red edge bands and broadband NIR.

Figure 9 illustrates the corresponding ExPAC model predicted TOA reflectance and EPICS-NA measurements for Sentinel-2A and Sentinel-2B, in all 11 spectral bands. The predicted TOA reflectance was plotted with respect to the TOA reflectance measurements; all bands were accumulated along the one-to-one line except the red edge bands: Band 5 and Band 7. Table 8 displays the statistical matrices: RMSE, Precision, and Accuracy, between the model predicted and measurements for Sentinel-2A and 2B using data from the beginning of life to May 2022. It should be noted that Sentinel-2 data, with Processing Version 4.00 introduced on 25 January 2022, was processed with offset, details in [44]. It was found that Red Edge 1 (Band 5) and Red Edge 3 (Band 7) had ~3–6% accuracy for both satellites. All other bands were at 2.5% accuracy or better, as seen in Table 8.

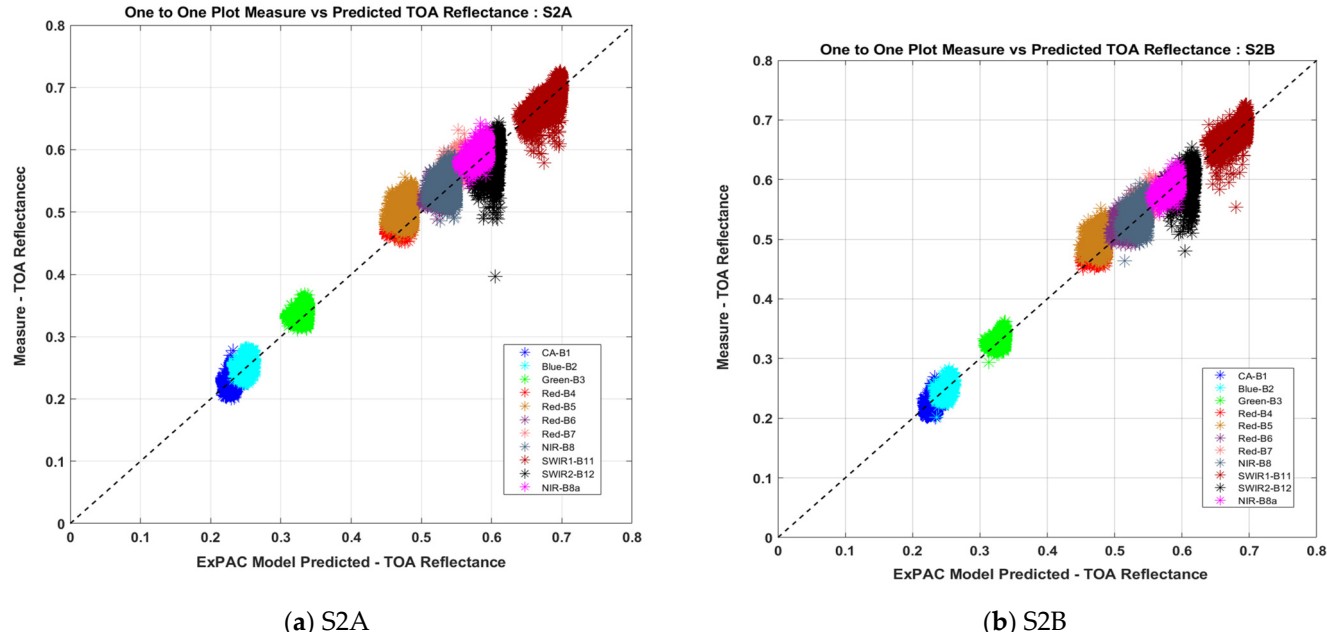

**(a)** S2A  **(b)** S2B

**Figure 9.** Comparison between the Expect Model results for Sentinel-2A (**a**) and Sentinel-2B (**b**) 11 spectral bands showing that the ExPAC Model predicted TOA Reflectance vs. measures.

**Table 8.** Statistical matrices showing Model Accuracy, RMSE and Precision between the ExPAC model predictions and Sentinel-2 measurements. The ExPAC Double Ratio between S2A and S2B is also included.

| Sentinel-2 Bands | Model Accuracy (%) | | RMSE (Reflectance Units) | | Precision (Reflectance Units) | | ExPAC D_Ratio |
|---|---|---|---|---|---|---|---|
| | S2A | S2B | S2A | S2B | S2A | S2B | S2A/S2B |
| CA | 1.92 | 0.33 | 0.010 | 0.010 | 0.013 | 0.010 | 1.016 |
| Blue | 0.77 | −0.21 | 0.010 | 0.010 | 0.010 | 0.010 | 1.009 |
| Green | −0.74 | −1.59 | 0.009 | 0.011 | 0.010 | 0.014 | 1.008 |
| Red | −1.11 | −2.10 | 0.012 | 0.016 | 0.016 | 0.024 | 1.011 |
| RE-1 | −4.52 | −5.59 | 0.027 | 0.032 | 0.048 | 0.059 | 1.010 |
| RE-2 | −1.27 | −1.27 | 0.016 | 0.015 | 0.020 | 0.019 | 1.000 |
| RE-3 | −2.93 | −3.92 | 0.020 | 0.025 | 0.035 | 0.046 | 1.011 |
| NIR | −0.81 | −1.32 | 0.016 | 0.018 | 0.018 | 0.022 | 1.006 |
| SWIR-1 | 0.03 | 0.56 | 0.015 | 0.015 | 0.015 | 0.017 | 0.994 |
| SWIR-2 | 2.01 | 2.25 | 0.023 | 0.025 | 0.031 | 0.034 | 0.998 |
| NIR-8A | −0.34 | −1.56 | 0.010 | 0.013 | 0.011 | 0.021 | 1.012 |

*5.2. The Application of the ExPAC Model*

5.2.1. The ExPAC Double Ratio

In order to perform the inter-comparison between two satellites using the ExPAC model, the ratio between the model predicted and the measurements (ExPAC Ratio) will be used. The ExPAC ratio between two satellites, referred to as the 'double ratio', is calculated to determine the differences for each spectral band. The double ratio allows direct comparison without potential method biases, as it applies practically and equally the same for both satellites.

The results are based on Landsat-8 and Landsat-9 data from November 2021 to March 2022 over EPICS-NA, comprising of 130 Landsat-8 images and 126 Landsat-9 images. The ratios were calculated using 7 days of co-incident pairs and filtering out ratios greater than 10%, so that the variation in sites due to different atmospheric conditions did not interfere with the analysis. The ExPAC Double Ratio results show that the two satellites are comparatively better than 0.5% for all bands except the green band with sub 1% agreement, as shown in Figure 10. The results also agreed with the Underfly event and other cross-calibration methods as reported by Gross et al. 2022 [11].

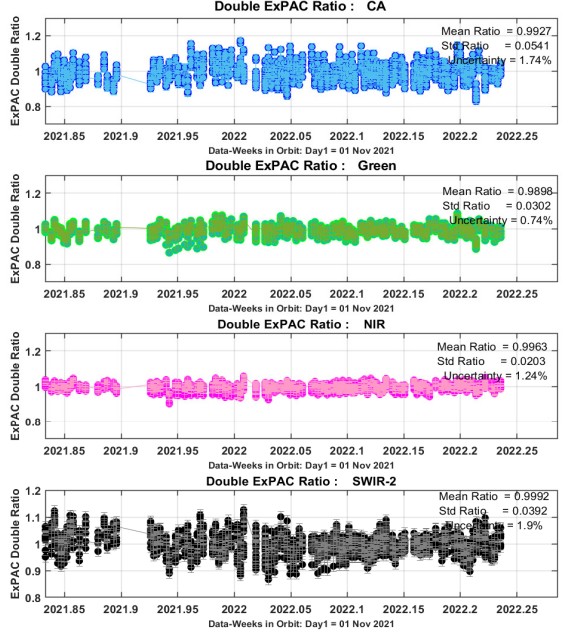
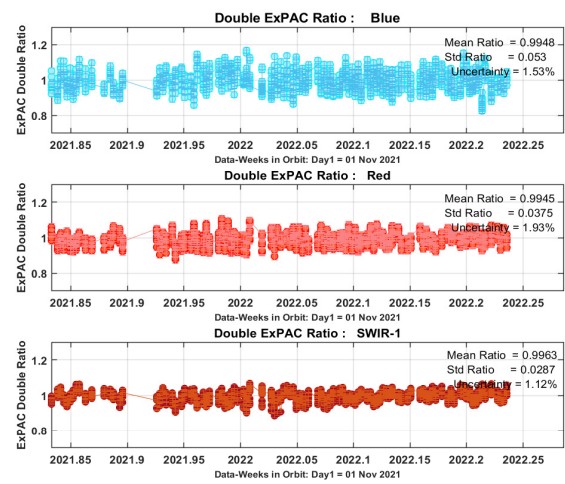

**Figure 10.** Landsat-8 vs. Landsat-9: ExPAC Double Ratio results during OIV period.

5.2.2. The ExPAC Double Ratio for Sentinel 2A–2B with 11 Bands and Inter-Comparison Results

The same calculations to determine ExPAC Double Ratio, as described in Section 5.2.1, were applied to Sentinel-2 MSI satellites, using Sentinel 2A (S2A) as a reference. The differences between these two satellites are within 1–2% for all 11 bands, as shown in Table 6. Even though the performance of the ExPAC Model for non-Landsat-8 spectral bands was not within 3% accuracy, using the double ratio has eliminated biases from the ExPAC Model because they are equally applied to S2A and S2B. Thus, the ExPAC Double Ratio provides consistent cross-calibration S2A-S2B results through all eleven spectral bands which show similar double ratio results of 1–2% compared with the other five techniques: Rayleigh and cross-calibration over four desert sites, as found in [32].

To improve the ExPAC model to accommodate non-Landsat-8 spectral bands, more information for these bands will be included to develop the BRDF model and the cross-calibration scale factor. This could be achieved by introducing Sentinel-2 data to form ExPAC data in the process, as described in Section 2.2.1.

*5.3. The Inter-Comparison of Landsat Missions vs. Sentinel-2 Missions Results*

5.3.1. The EPICS-Based Cross-Calibration Results

This section presents the inter-comparison results of the Landsat missions and the Sentinel-2 missions using the ExPAC Double Ratio and Traditional EPICS-Based Cross-Calibration Ratio (EPICS-NA, EPICS-Global). Satellite data used for this study were from the beginning of its life until August 2022, detailed in Table A3. Each method provided the cross-calibration ratio for all seven spectral bands with agreement well within 1–2% associated with uncertainties derived in Section 4, as shown in Figures 11–13. However, the

inter-comparison of Landsat-9 and Landsat-7 using data from November 2021 to August 2022 (L9 images: EPICS-NA = 287, EPICS-Global = 420, L7 images: EPICS-NA = 122, EPICS-Global = 384) images showed differences of approximately 1–4% for all three methods. It should be noted that Landsat-7 has been placed in the lowering orbit since mid-2021 to allow Landsat-9 to take its orbital place at 805 km. In April 2022, Landsat-7 had been placed into the lower orbit at 697 km [26].

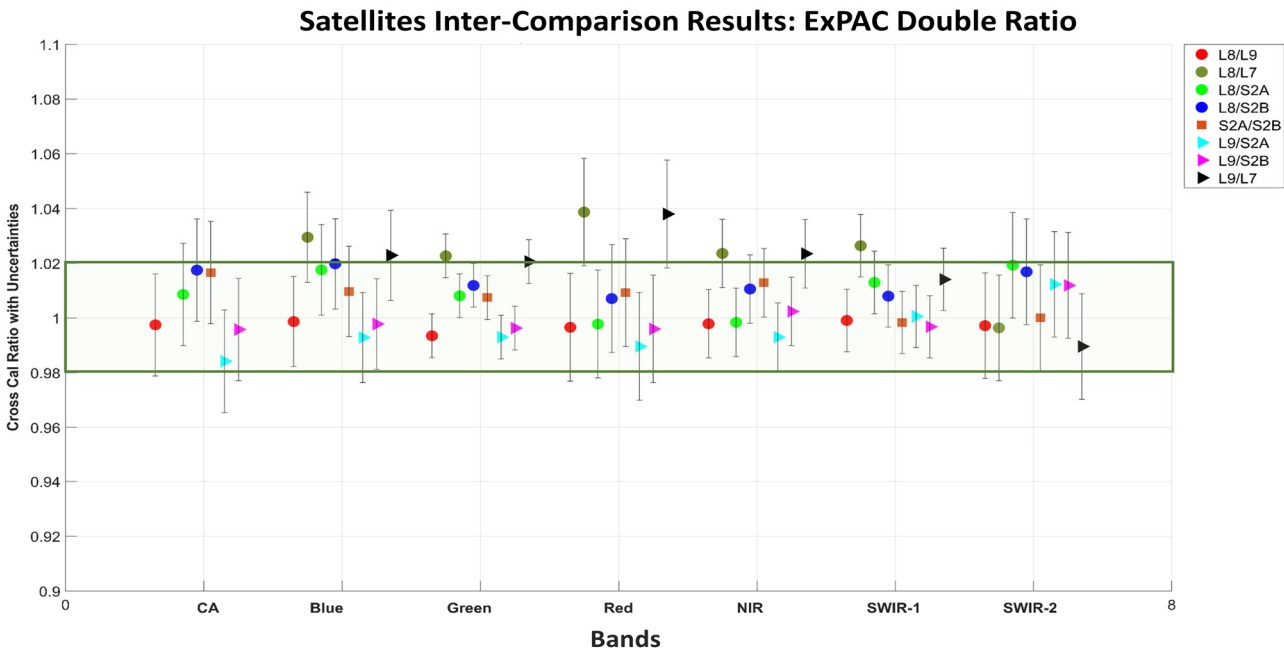

**Figure 11.** ExPAC Double Ratio for Inter-comparison of Landsats and Sentinel-2(s). The green box highlights the cross-calibration ratio agreement within 2%.

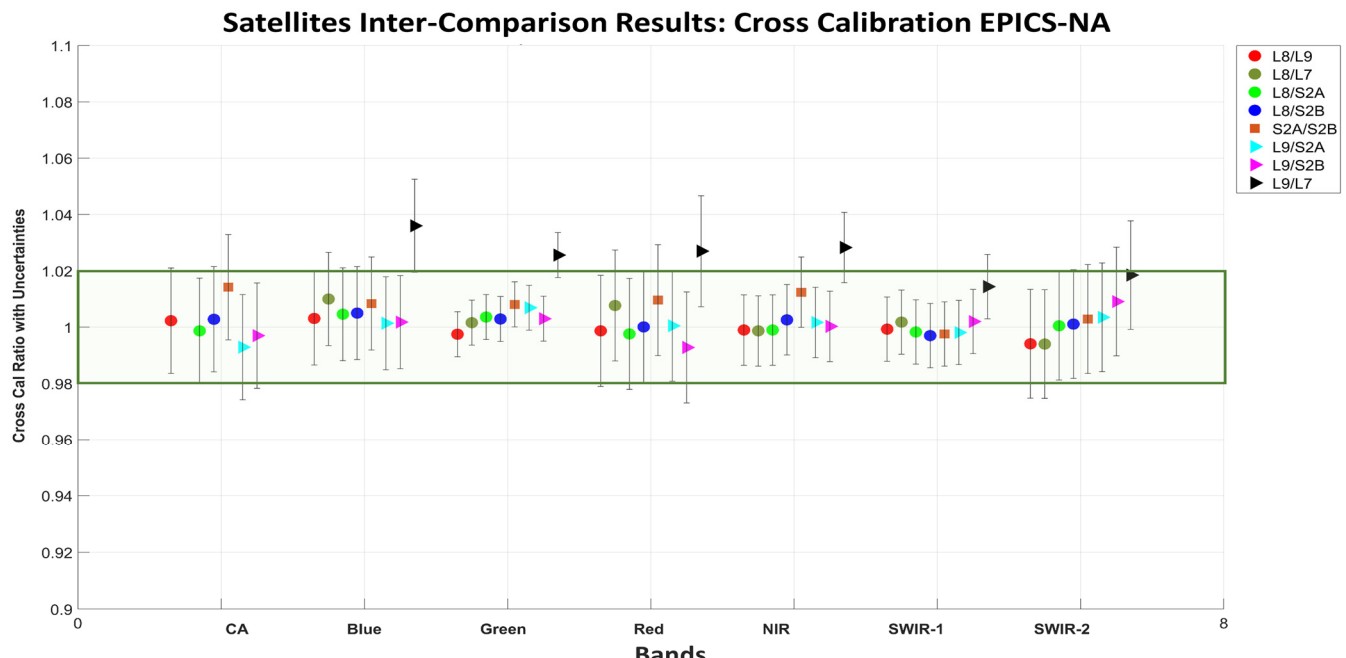

**Figure 12.** EPICS-based cross-calibration EPICS-NA Ratio for Inter-comparison of Landsats and Sentinel-2(s). The green box highlights the cross-calibration ratio agreement within 2%.

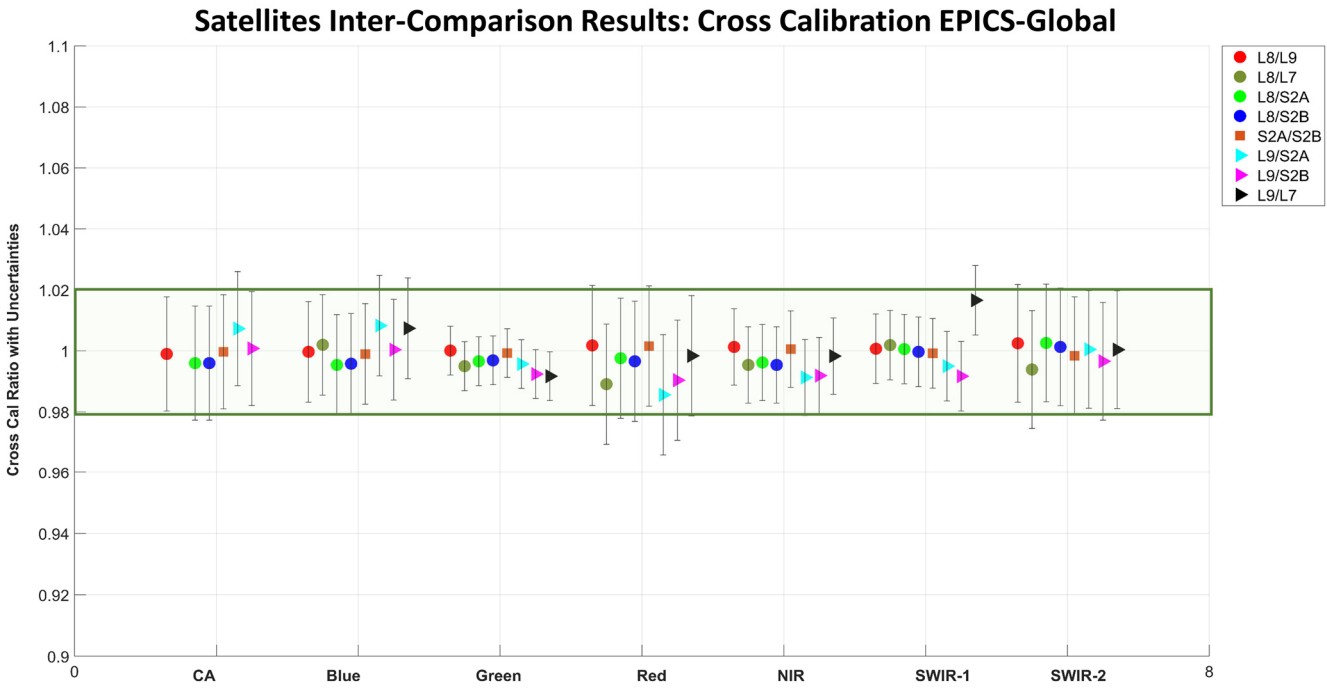

**Figure 13.** EPICS-based cross-calibration EPICS-Global Ratio for Inter-comparison of Landsats and Sentinel-2(s). The green box highlights cross-calibration ratio agreement within 2%.

5.3.2. The Inter-Comparison Landsat-8 vs. Landsat-9 during OIV with the Three Advanced Methods

With three distinctive approaches, the results show that the two satellites agree well within 0.5% for all spectral bands; only the green band shows sub 1% difference, as shown in Figure 14 and Table 8. The uncertainties were at a level of 1–3% for all spectral bands.

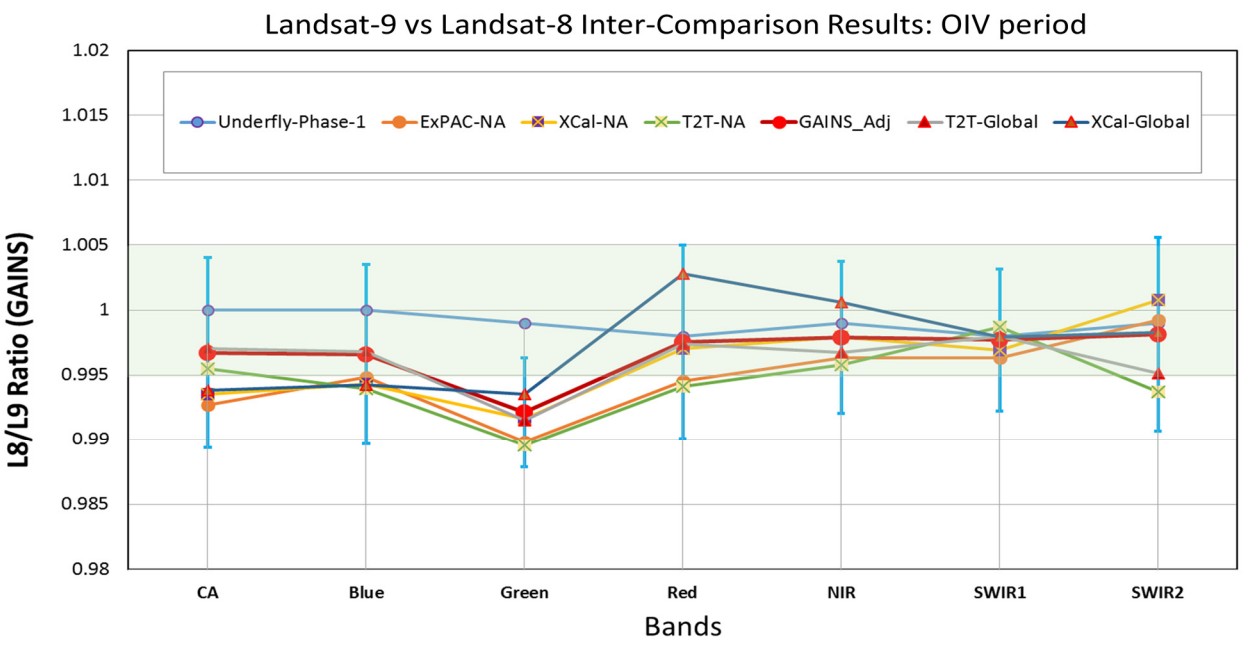

**Figure 14.** SDSU IPLab inter-comparison results with multi validation approaches: Underfly-Phase 1 (blue), Trend to Trend with EPICS-NA (green line) and EPICS-Global (grey line). This paper shows ExPAC Double Ratio (ExPAC-NA, orange line), EPICS-based Cross-Calibration Ratio, with EPICS-NA (XCal-NA, yellow line), EPICS-Global (XCal-Global, dark blue line), and Gains Adjustment (red line).

The SDSU Image Processing Lab is a member of the Landsat Cal/Val Team who provided multiple approaches for assessing the performance of Landsat-9 OIV, namely, Underfly Event, EPICS Trend-to-Trend (T2T), ExPAC Double Ratio, and EPICS-based cross-calibration [11]. All these results were reported to be within 0.5–1% for all spectral bands confirming that the EPICS-based cross-calibration methods and ExPAC Double Ratio were comparable and successful methods, as seen in Figures 11–14.

Utilizing EPICS-NA for an extended EPICS absolute calibration model (ExPAC model) and using it as a tool to perform cross-calibration, the ExPAC Double Ratio can provide the same level of agreement similar to other methods, as seen in Table 9 and Figure 14. Hence, the EPICS-based cross-calibration techniques and the ExPAC Double Ratio can routinely be used as a tool to perform cross-calibration between two satellites with the capability of daily observation owing to the global coverage of EPICS-Global and EPICS-NA.

**Table 9.** Summary of the ExPAC Double Ratio and the EPICS-based Cross-Calibration Ratio: Landsat-8 vs. Landsat-9 during OIV.

| GAINS | The EPICS-Based Cross-Calibration Ratio and the ExPAC Double Ratio Results | | | | | | |
|---|---|---|---|---|---|---|---|
| | CA | Blue | Green | Red | NIR | SWIR-1 | SWIR-2 |
| EPICS-NA | 0.994 | 0.994 | 0.992 | 0.997 | 0.998 | 0.997 | 1.001 |
| Std. Dev | 0.048 | 0.046 | 0.029 | 0.037 | 0.021 | 0.028 | 0.039 |
| Uncertainty (%) | 2.21% | 2.03% | 1.38% | 1.44% | 1.09% | 1.27% | 1.74% |
| EPICS-Global | 0.994 | 0.994 | 0.994 | 1.003 | 1.001 | 0.998 | 0.998 |
| Std. Dev. | 0.049 | 0.052 | 0.035 | 0.051 | 0.034 | 0.033 | 0.046 |
| Uncertainty (%) | 1.66% | 1.58% | 1.20% | 1.33% | 1.01% | 1.26% | 1.53% |
| ExPAC D Ratio | 0.993 | 0.995 | 0.999 | 0.995 | 0.996 | 0.996 | 0.999 |
| Std. Dev. | 0.054 | 0.053 | 0.030 | 0.038 | 0.020 | 0.029 | 0.039 |
| Uncertainty (%) | 1.87% | 1.65% | 0.80% | 1.97% | 1.25% | 1.14% | 1.93% |

To reinstate the capability of the three advanced methods with respect to existing cross comparison results between Landsat-8 and Sentinel-2A, they consistently agreed within 1–2% for all seven spectral bands [1,33,45]. The differences between Sentinel-2A and 2B were reported to be less than 1.5% for all eleven bands, as shown in Table 8 and Figures 11–13; similar results can be found in [42,45,46]. With the use of EPICS-Global as the target, the opportunity to acquire calibration on a daily basis is possible and the outcome is reliable and comparable to other cross-calibration methods.

## 6. Conclusions

EPICS-based cross-calibration and ExPAC Double Ratio approaches are proven to be compatible tools for validating the performance of Landsat-9 against Landsat-8 during OIV. The results provided in this paper were generated using 5 months of data (November 2021 to March 2022). They provided cross-calibration gain ratios between these two sensors at 0.5–1% difference levels with uncertainties of 1–2.2% for all bands. The capability of EPICS-NA and EPICS-Global for satellite radiometric performance assessment provides a daily calibration opportunity. Using these two targets also enables an independent cross check between methodologies, as the expected results should agree well within 0.5%, as shown in Table 9 and Figure 14. It is also attractive to any sensors, especially short-lived small/nanosatellites, with no on-board calibrator to monitor the health and performance of satellites in a short period of time. It has been demonstrated that the uncertainties can be achieved at 2% levels with 25 weeks of data for the EPICS-based cross-calibration and the ExPAC Double Ratio.

The ExPAC Double Ratio is capable of performing cross-calibration gain ratios in Sentinel-2/MSI 11 spectral bands with 1–2% agreement level for all bands. However, the performance of the ExPAC Model with red edge bands is not satisfactory as the accuracy was at the level of 3–5%. Therefore, there will be an improvement in developing the ExPAC

model for non-Landsat-8 equivalent bands. The existing ExPAC Model (EPICS-NA as a target) can also be expanded to EPICS-Global with some additional red edge bands.

Gross et al. [11] stated that Underfly Gains estimated were not taken into account when examining spectral band differences, the green band showing largest different result than the other three independent methods: the Trend to Trend(EPICS-NA, EPICS-Global), the EPICS-based cross-calibration (EPICS-NA, EPICS-Global), and the ExPAC Double Ratio, as shown in Figure 14. After combining results from all six methods, the gain adjusted for the green band was sub 1%, whereas that for all other bands was ~0.5%. However, more analysis is still being carried out for the four independent methods and will be used to support decision making to update Landsat-9 calibration gains in the near future.

**Author Contributions:** Conceptualization, M.K.; Formal analysis, M.K.; Funding acquisition, L.L.; Methodology, M.K.; Resources, L.L., R.S. and G.G.; Supervision, L.L.; Validation, M.K.; L.L., R.S. and G.G.; Writing—original draft, M.K.; Writing—review and editing, M.K. and L.L. All authors have read and agreed to the published version of the manuscript.

**Funding:** This research was funded by National Aeronautics and Space Administration Radiometric Calibration grant number SA22000091 from the Landsat Project Science Office; the United States Geological Survey/EROS Landsat 8-9 grant number SA2000371.

**Data Availability Statement:** Landsat-8 and Landsat-9 courtesy of the US Geological Survey Collection 2 Landsat 8-9 OLI/TIR Digital Object Identifier (DOI) number:/10.5066/P975CC9B.

**Acknowledgments:** The authors would like to thank the Landsat Calibration/Validation Team, USGS EROS Data Center, NASA Goddard Space Flight Center and the members of South Dakota State University Image Processing Laboratory for their support and helpful suggestion for this project.

**Conflicts of Interest:** The authors declare no conflict of interest.

## Appendix A

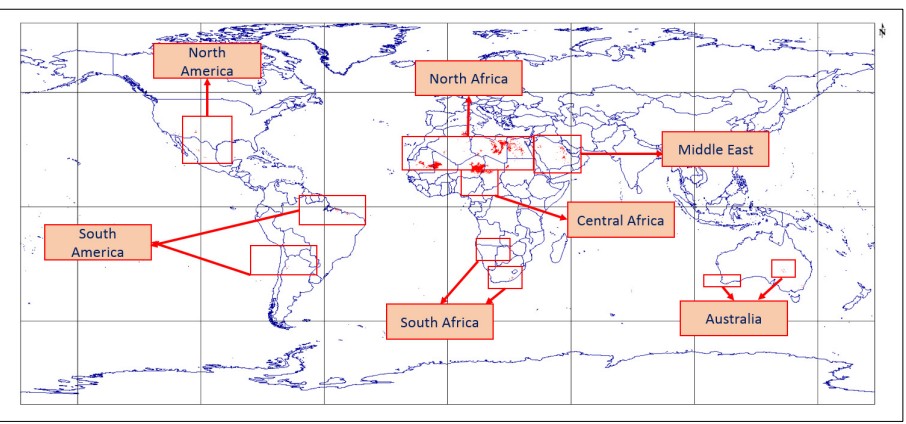

**Figure A1.** The footprints of EPICS-Global, total of 33 WRS Path/Row(s).

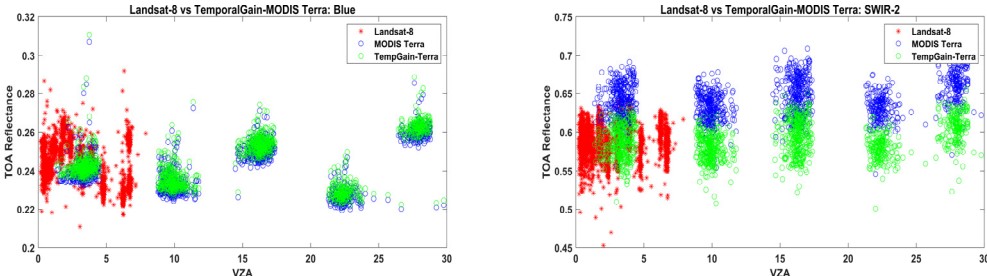

**Figure A2.** Before and after applying Temporal Gain MODIS Terra and Landsat-8 data versus view zenith angles.

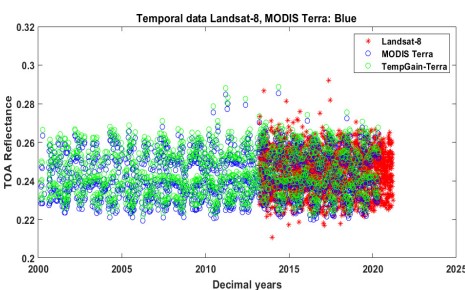 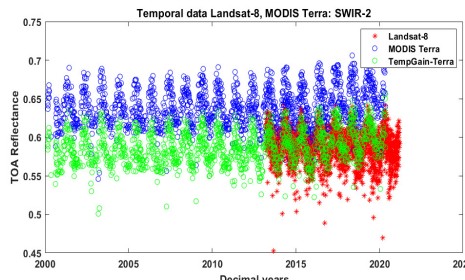

**Figure A3.** Before and after applying Temporal Gain MODIS Terra and Landsat-8 data: Temporal data.

**Table A1.** The results of 'The Wilcoxon Rank Sum Test' for all 6 satellites, to test if the satellite measurements and the ExPAC predicted TOA reflectance have the same medians (h = 0).

| | The Wilcoxon Rank Sum Test (*p*-Value) | | | | | | The Wilcoxon Rank Sum Test (h) | | | | | |
|---|---|---|---|---|---|---|---|---|---|---|---|---|
| | Terra | Aqua | L7 | L8 | S2A | S2B | Terra | Aqua | L7 | L8 | S2A | S2B |
| CA | – | – | – | 0.087 | 0.085 | 0.456 | – | – | – | 0 | 0 | 0 |
| Blue | 0.094 | 0.084 | 0.084 | 0.373 | 0.097 | 0.849 | 0 | 0 | 0 | 0 | 0 | 0 |
| Green | 0.100 | 0.086 | 0.277 | 0.317 | 0.238 | 0.777 | 0 | 0 | 0 | 0 | 0 | 0 |
| Red | 0.085 | 0.193 | 0.412 | 0.673 | 0.416 | 0.411 | 0 | 0 | 0 | 0 | 0 | 0 |
| NIR | 0.091 | 0.305 | 0.736 | 0.616 | 0.090 | 0.139 | 0 | 0 | 0 | 0 | 0 | 0 |
| SWIR-1 | 0.870 | – | 0.305 | 0.594 | 0.460 | 0.110 | 0 | – | 0 | 0 | 0 | 0 |
| SWIR-2 | 0.921 | 0.398 | 0.340 | 0.217 | 0.422 | 0.341 | 0 | 0 | 0 | 0 | 0 | 0 |

**Table A2.** Student's *t*-test and the hypothesis test result: 15 BRDF coefficients for the ExPAC model development: SWIR-1 band.

| Coefficient | Estimate | Standard Error | T-Statistics | *p*-Value Statistical | Statistical Response |
|---|---|---|---|---|---|
| Intercept | 0.7005 | 0.0003 | 2361.149 | 0 | Significant |
| $Y_1$ | 0.0000 | 0.0003 | <0 | 1 | Insignificant |
| $X_1$ | 0.0000 | 0.0003 | <0 | 1 | Insignificant |
| $Y_2$ | 0.0000 | 0.0037 | <0 | 1 | Insignificant |
| $X_2$ | 0.0000 | 0.0006 | <0 | 1 | Insignificant |
| $X_1Y_1$ | 0.0000 | 0.0008 | <0 | 1 | Insignificant |
| $Y_1Y_2$ | 0.1754 | 0.0155 | 11.293 | <0 | Significant |
| $X_1Y_2$ | 0.0000 | 0.0181 | <0 | 1 | Insignificant |
| $X_2Y_1$ | 0.0000 | 0.0026 | <0 | 1 | Insignificant |
| $X_1X_2$ | 0.0094 | 0.0029 | 3.2460 | 0.001 | Significant |
| $X_2Y_2$ | 0.0000 | 0.0117 | <0 | 1 | Insignificant |
| $Y_1^2$ | −0.0727 | 0.0006 | −117.254 | 0 | Significant |
| $X_1^2$ | −0.0588 | 0.0015 | −40.519 | 0 | Significant |
| $Y_2^2$ | −1.9655 | 0.1240 | −15.845 | <0 | Significant |
| $X_2^2$ | 0.0722 | 0.0038 | 19.183 | <0 | Significant |

**Table A3.** Data used for inter-comparison of satellites: EPICS-NA and EPICS-Global.

| Satellites | No. of Images | | |
|---|---|---|---|
| | EPICS-NA | EPICS-Global | Data Till |
| Landsat-7 | 2168 | 7510 | 31 August 2022 |
| Landsat-8 | 2751 | 4910 | 31 August 2022 |
| Landsat-9 | 266 | 420 | 31 August 2022 |
| Sentinel-2A | 3329 | 5113 | 31 August 2022 |
| Sentinel-2B | 2484 | 3495 | 31 August 2022 |

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
