# Peer review of "Inter-Comparison of Landsat-8 and Landsat-9 during On-Orbit Initialization and Verification (OIV) Using Extended Pseudo Invariant Calibration Sites (EPICS): Advanced Methods"

_remotesensing, doi:10.3390/rs15092330_

Round 1
Reviewer 1 Report
The study was solid work. However, I do suggest several edits.
1.Line 13, “ExPAC” is abbreviated, please give the complete words. In fact, there are a lot of abbreviated terms in the abstract part, please rewrite.
2.Line 20, explain ” The EPICS-based inter-comparison uncertainties were determined using data on a weekly basis; analysis with 25 weeks of data and Monte Carlo Simulation.”
3. Line 51. “This paper focused on three newly developed methods”, Traditional Extended PICS-based cross calibration cannot be considered as a newly developed method, please explain.
4. This paper focused on landsat-8 OLI? or both OLI and TIRS?
5. Line 270, why could all hyperspectral profiles be referred as “Calibrated Hyperspectral profile”. All hyperspectral profiles should be described.
6.Line 292, please provide more information about 4 Angle BRDF Model.
7.Line 386, viewing angle? zenith angle or azimuth angle?
8.Line 517, “ExPAC Double Ratio Uncertainties Analysis”, there was no content related to Uncertainties Analysis.
9.Line 649, Why was Landsat-7 data using in this part?
10. In conjunction with the above, I feel you are definitely using too many significant figures in your paper (I must admit that I am a stickler for significant figures and what they imply). I think every table and figure you used should be cited in your text and explained in detail.
Reviewer 2 Report
In this manuscript, several work on Inter-comparison of Landsat-8 and Landsat-9 during On-Orbit Initialization and Verification (OIV) Using Extended Pseudo Invariant Calibration Sites (EPICS) is presented. I think this study is valuable and meaningful. In particular, using pseudo invariant sites for sensor performance monitoring and cross calibration is an important research field in radiometric calibration. However, there are some issues in this manuscript that need to be addressed in order to achieve publication level.
(1) Some unclear statements need to be modified and supplemented, such as:
(a) In section 2.2.1, the temporal gain to bring Hyperion to match Landsat-8 spectral band was introduced, but that for Terra to L8 isn’t presented. I am concerned about this because the view angles of Hyperion and Landsat 8 are both small, and the impact of angle differences can be ignored when calculating temporal gains. However, for Terra, the view angle is large. When calculating temporal gains, how to deal with the view angle differences between Terra and Landsat 8 is important, and it is necessary to provide additional information in the manuscript.
(b) I don't understand Table 3 very well, because from the ExPAC dataset, it can be found that the CV (%) values are very small. However, it is said in the manuscript that TOA reflectance data from different view angles are used for generating 4 angle BRDF model. So, whether the impact of view angles on TOA reflectance is not significant or whether the statistics in Table 3 are limited by view angle differences.
(c) There is no explanation for Equation (16), which of the formulas is the reference TOA reflectance?
(d) In the abstract, the authors states that the development of these two techniques were described with uncertainties analysis. However, only the uncertainty analysis associated with the ExPAC Double Ratio is presented in this manuscript.
(2) The description of methods and results is somewhat confusing, for example, sections 2.3, 2.4, and 6.1 are all validation of the ExPAC model. For better readability, I hope to reorganize the structure of the manuscript.
(3) Some minor writing issues are as follows.
(a) I think that the first time an abbreviation appears in a abstract, it is necessary to give it a full name like “initialization and verification (OIV)”.
(b) In line 24, “Lands4at-8” should be Landsat-8.
(c) In line 45, what’s the “CPF”?
(d) Figure 4 is a combination of a graph and a table, which is not common. And the quality of Figure 4 is poor and needs to be replaced with a higher quality figure. And so is Figure 6.
Reviewer 3 Report
All the comment is attached in the pdf file

Round 2
Reviewer 2 Report
There are no new comments.